# Simultaneous triple-parametric optical mapping of transmembrane potential, intracellular calcium and NADH for cardiac physiology assessment

Sharon A. George [1,2,3✉], Zexu Lin[1] & Igor R. Efimov [1,2,3✉]

Investigation of the complex relationships and dependencies of multiple cellular processes that govern cardiac physiology and pathophysiology requires simultaneous dynamic assessment of multiple parameters. In this study, we introduce triple-parametric optical mapping to simultaneously image metabolism, electrical excitation, and calcium signaling from the same field of view and demonstrate its application in the field of drug testing and cardiovascular research. We applied this metabolism-excitation-contraction coupling (MECC) methodology to test the effects of blebbistatin, 4-aminopyridine and verapamil on cardiac physiology. While blebbistatin and 4-aminopyridine alter multiple aspects of cardiac function suggesting off-target effects, the effects of verapamil were on-target and it altered only one of ten tested parameters. Triple-parametric optical mapping was also applied during ischemia and reperfusion; and we identified that metabolic changes precede the effects of ischemia on cardiac electrophysiology.

[1] Department of Biomedical Engineering, The George Washington University, Washington, DC, USA. [2] Department of Biomedical Engineering, Northwestern University, Chicago, IL, USA. [3] These authors jointly supervised this work: Sharon A George, Igor R Efimov. ✉email: sharonann.george@northwestern.edu; igor.efimov@northwestern.edu

Cardiac physiology is profoundly complex, and the careful coordination of numerous cell signaling pathways governs every single heartbeat[1,2]. These physiological processes that vary beat to beat and long term are usually sub-divided into metabolic, electrical, and mechanical components governed by metabolism-excitation-contraction coupling (MECC). Electrical activity in the heart includes a sequence of opening and closing of ion channels and pumps which cause cardiomyocyte depolarization and repolarization. The resulting changes in the transmembrane potential ($V_m$) in the cardiomyocytes are recorded as action potentials using electrical or optical methods. The electrical excitation of the heart serves as the trigger of a mechanical contraction. Excitation-contraction coupling or the translation of electrical excitation to mechanical contraction is controlled by cytosolic calcium ($Ca^{2+}$) ion concentration, which increases following electrical excitation and $Ca^{2+}$ ion binds to a contractile protein in the cardiomyocyte, triggering contraction. Both the electrical and mechanical processes of the heart require energy, which is provided in the form of ATP generated by metabolic processes in the mitochondria. Thus, all three components of cardiac function are intertwined into MECC. As such, studying this complex MECC phenomenon requires the ability to simultaneously assess these three facets of cardiac function. In this study, we report a new approach to simultaneously image $V_m$, $Ca^{2+}$, and NADH (metabolic marker).

Optical mapping is a methodology that optically records cardiac physiology with a high spatial and temporal resolution, either as autofluorescence of endogenous biological substances or as fluorescence of specifically designed dyes[3,4]. Optical mapping of the heart was first applied to record $V_m$ and then NADH autofluorescence[5,6]. Since then, optical mapping has also been applied to record $Ca^{2+}$[7]. Dual parameter optical mapping of $V_m$ and NADH, as well as $V_m$ and $Ca^{2+}$, have also been applied in assessing cardiac physiology[8,9]. However, as described above, the three interdependent facets of cardiac function will all need to be measured simultaneously to develop a complete picture of cardiac physiological modulation by drugs or disease. In this study, we report for the first time, a spatially and temporally co-registered triple-parametric optical mapping system that incorporates three cameras to simultaneously capture NADH, $V_m$, and $Ca^{2+}$ signals from the same field of view.

Preclinical safety and efficacy testing are crucial components of the drug development process. This step is important in determining dosing and toxicity which could include identifying off-target effects of drugs before clinical trials and before they are approved for use in patients. Cardiotoxicity is the primary cause (19%) of drug withdrawal from the market in the United States[10] and the second leading cause worldwide[10,11], underscoring the need for efficient and thorough cardiac methodologies of drug screening. Current preclinical drug testing primarily focuses on the effects of drugs on the electrical activity (QT interval) and contractility of the heart[10,12]. While this is an essential first step, it does not give a complete picture of cardiac physiology modulation by drugs as it does not consider the calcium handling or metabolic states of the cardiac tissue. Unexpected off-target effects of the drugs being tested could result in serious complications or fatality. In this study, we present a novel approach to measure ten different important aspects of cardiac MECC using triple-parametric optical mapping. We investigated the effects of three different compounds (blebbistatin, 4-aminopyridine, and verapamil) using triple-parametric optical mapping and present the data in a ten-parameter panel (TPP). TPP graphs include information on action potential upstroke, duration and conduction, intracellular calcium release, and reuptake as well as the metabolic state of the heart.

Triple-parametric optical mapping and TPP graphs could also benefit the study of complex cardiac diseases such as ischemia and reperfusion. Acute ischemic bouts are known to have multiple and severe effects on cardiac physiology. Ischemia has been previously demonstrated to alter the electrical activity[13–15], calcium handling[16] as well as the metabolic[6] functions of the heart. However, the sequence and the interrelationship between these three aspects of cardiac MECC have not been studied simultaneously before, due to the lack of appropriate methodology. In this study, we also determined the simultaneous modulation of multiple aspects of cardiac physiology by ischemia and their restoration during reperfusion. Thus, applying triple-parametric optical mapping to study MECC during disease progression could provide valuable new targets for therapy.

## Results

Triple-parametric optical mapping system was 3D printed and set up as illustrated in Fig. 1a, b and the separation of signals of different wavelengths is illustrated in Fig. 1c. All design files for 3D printed hardware (in STL format) and data analysis software (Matlab) are available under an open-source license at Github (https://github.com/optocardiography, DOI: 10.5281/zenodo.5784023). The following ten parameters were measured from the optical recordings and an illustration of each parameter definition is included in Fig. 1d–f. From the NADH recordings, the absolute intensity of the NADH signals was measured which corresponds to NADH concentration in the tissue. Since this parameter can vary between hearts depending on experimental conditions, all measurements from a given heart were normalized to the first recording from that same heart (Control at 200 ms pacing rate for drug testing protocol and Baseline at 200 ms pacing rate for ischemia protocol). This allows the determination of changes in NADH induced by drug treatment or disease without confounding interexperimental variables. From the depolarization/calcium release phase of the $V_m$ and Ca signals, $V_m$ and Ca rise times ($V_m$ RT and Ca RT), longitudinal and transverse conduction velocity ($CV_L$ and $CV_T$), anisotropic ratio (AR), and activation delay between $V_m$ and Ca traces ($V_m$-Ca delay) were calculated. Rise time was defined as the time taken for depolarization, from 20 to 90% of signal upstroke. In the case of $V_m$ RT, this parameter indicates function of depolarizing currents while in the case of Ca RT, this parameter indicates the time taken for calcium entry into the cell and calcium-induced calcium release from the sarcoplasmic reticulum. Conduction velocity is the speed with which the activation wavefront travels in a given direction, longitudinal (parallel to fiber orientation) and transverse (perpendicular to fiber orientation). AR is the ratio of $CV_L$ to $CV_T$ and indicates the ellipticity of the propagating wavefront. Higher AR is associated with increased arrhythmogenicity. $V_m$-Ca delay is calculated to determine the excitation-contraction coupling. Prolonged delay suggests uncoupling between electrical excitation and mechanical contraction. From the repolarization/calcium reuptake phase of the $V_m$ and Ca signals, action potential duration at 80% repolarization ($APD_{80}$), calcium transient duration at 80% reuptake ($CaTD_{80}$), and calcium decay constant (Ca $\tau$) were calculated. It is important to note that these three parameters ($APD_{80}$, $CaTD_{80}$, and Ca $\tau$) were only measurable in hearts after Blebbistatin perfusion. Without Blebbistatin, motion artifacts were present in the signals which distort the $V_m$ and Ca signals in the repolarization/reuptake phase (Fig. 2a, top). $APD_{80}$ was defined as the time interval between activation and repolarization or in other words the duration for one cardiac cycle. Similarly, $CaTD_{80}$ was defined as the time interval between calcium release and reuptake of calcium back into the sarcoplasmic reticulum. Both shortening and prolongation of these parameters have been reported to be arrhythmogenic. Lastly, Ca $\tau$ is the decay constant measured by fitting an exponential to the reuptake phase of the calcium transients. This parameter is indicative of how quickly calcium in the cytoplasm is removed after each contraction.

**Blebbistatin modulates cardiac physiology.** Motion of the heart during image acquisition introduces artifacts in the recorded optical

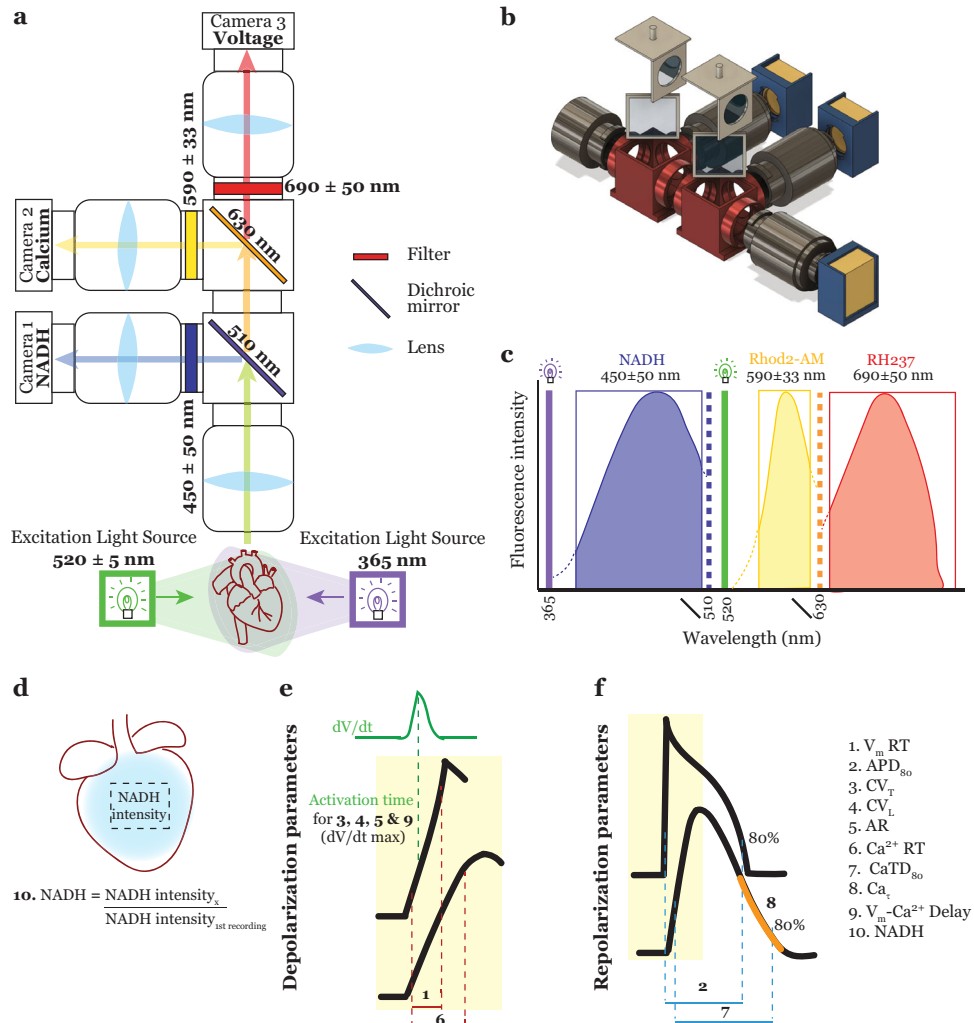

**Fig. 1 Triple-parametric optical mapping system. a** Schematic of the triple-parametric optical mapping system illustrating the optics used in the system and the light path for the three signals—$V_m$, $Ca^{2+}$, and NADH. **b** 3D rendering of the triple parameter optical mapping system illustrating the 3D printed hardware that houses the optics used in the system. **c** Spectra of the three parameter signals—NADH autofluorescence, Rhod2-AM ($Ca^{2+}$), and RH237 ($V_m$) fluorescence, illustrating the separation of the three signals as implemented in this system. Solid vertical lines: LED light source wavelength, dotted vertical lines: dichroic mirror, boxes: filters. Illustration of the different parameters measured using the three optical signals. **d** NADH intensity was determined and each intensity value from a given heart was normalized to the NADH intensity from the first recording from that heart. **e** Definition of parameters measured from the depolarization phase of the $V_m$ signal or the calcium release phase of the $Ca^{2+}$ signal. **f** Definition of the parameters measured from the repolarization phase of the $V_m$ signal or the calcium reuptake phase of the $Ca^{2+}$ signal.

signals which hinder the analysis of repolarization-related parameters. To prevent these motion artifacts, electromechanical uncouplers such as blebbistatin are routinely used in the optical mapping of $V_m$ and $Ca^{2+}$. Thus, the first step in this study was to analyze the effects of blebbistatin on the ten parameters of cardiac physiology.

Representative $V_m$ and $Ca^{2+}$ traces (Fig. 2a top) illustrate motion artifacts that are introduced in the repolarization phases in Control (no treatment) hearts. Activation/intensity maps generated from these traces are shown in Fig. 2b (top). Optical recordings during Control treatment allowed for the measurement of seven parameters –$V_m$ RT, $CV_T$, $CV_L$, AR, $Ca^{2+}$ RT, $V_m$-$Ca^{2+}$ delay, and NADH. Restitution, which is the property of electrophysiological parameters to vary with diastolic interval (typically decrease with decreasing diastolic interval), was observed in $CV_T$ and $CV_L$.

Treatment with blebbistatin (15 μM) abolished contractions and removed motion artifacts as shown in Fig. 2a (bottom). Preventing motion-induced distortion of signals in the later phases of the action potential and calcium transient allowed the

measurement of repolarization/calcium reuptake parameters such as $APD_{80}$, $CaTD_{80}$ and Ca τ. Restitution property was observed in $APD_{80}$, $CV_T$, and $CV_L$. Additionally, $CaTD_{80}$ and $Ca_\tau$ was also rate dependent. Specifically, both $CaTD_{80}$ and $Ca_\tau$ decreased with increasing pacing rate (Fig. 2c).

Blebbistatin also induced significant differences in cardiac physiology compared to Control. Blebbistatin increased both $V_m$ and $Ca^{2+}$ RTs ($p = 0.005$ and $0.002$, respectively). On the other hand, NADH intensity was reduced after blebbistatin treatment ($p < 0.001$, Fig. 2c).

In the TPP graph in Fig. 2d, percent change in each of the ten parameters induced by blebbistatin with respect to Control, at 150 ms BCL (basic cycle length), is summarized which once again illustrates that blebbistatin significantly increases $V_m$ and $Ca^{2+}$ RTs ($p = 0.002$ and $0.005$, respectively).

**On- and off-target effects of drugs on cardiac physiology.** The application of triple-parametric optical mapping in drug testing

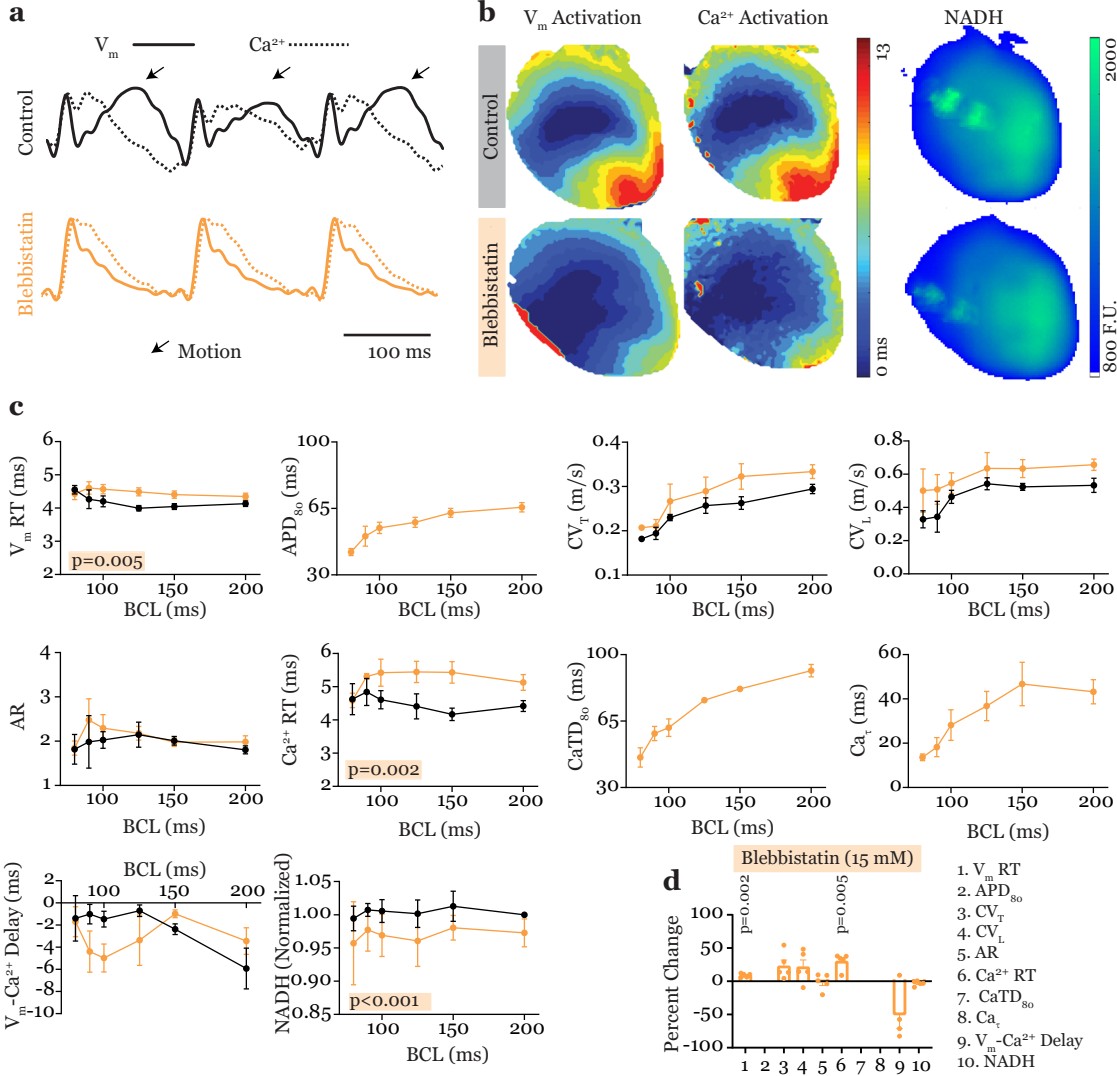

**Fig. 2 Electromechanical uncoupling in triple-parametric optical mapping. a** Representative traces of $V_m$ and $Ca^{2+}$ (solid and dotted traces, respectively) were recorded without (Control, black) and with the electromechanical uncoupler Blebbistatin (15 μM, orange). **b** Representative $V_m$ activation maps (left), $Ca^{2+}$ activation maps (middle), and NADH intensity maps (right) recorded during Control and Blebbistatin treatment. All maps were recorded from the same heart, the top three maps were from simultaneous recording during Control and the bottom three from simultaneous recording during Blebbistatin treatment. Slightly different silhouettes, particularly around the boundaries is due to different background noise removal levels using a thresholding algorithm. These boundary pixels were not used in the analysis. **c** Summary restitution properties of the ten parameters measured simultaneously by triple-parametric optical mapping. **d** Ten-parameter panel showing the effects of Blebbistatin on cardiac physiology at 150 ms BCL, reported as percent change from Control. Data presented as mean ± s.e.m., n = 5 hearts, Least squares regression analysis was performed for data in panel (**c**) to detect significant differences in parameters during Control versus Blebbistatin treatment and all significant p values are reported. Two-tailed, paired t-tests were performed for data in panel (**d**) to detect significant change induced by Blebbistatin compared to Control and all significant p values are reported. Benjamini–Hochberg correction was applied to account for multiple comparisons.

was then evaluated with two well-studied drugs currently used in treating patients—4-AP and verapamil. The effects of these drugs on cardiac physiology were tested in the presence of blebbistatin to be able to determine repolarization-/calcium reuptake-related parameters. Therefore, each physiological parameter during 4-AP and verapamil treatment were compared to blebbistatin treatment to determine significant drug-related effects. The effects of 4-AP and verapamil on cardiac physiology are summarized in Fig. 3 and in Supplementary Data 1. The TPP graphs in Fig. 3a, demonstrate the differences between 4-AP and verapamil. While 4-AP had multiple on-target and off-target effects on cardiac physiology, verapamil only had a specific on-target effect.

*Effects of 4-AP on cardiac physiology.* Treatment with 4-AP (7 mM), a transient outward potassium current ($I_{to}$) blocker, prolonged $APD_{80}$ ($p < 0.001$) at all tested pacing rates compared to blebbistatin, as expected. 4-AP treatment also prevented pacing the hearts at pacing rates faster than 125 ms BCL. Additionally, at 150 ms BCL, 4-AP prolonged $Ca^{2+}$ RT ($p < 0.024$) and slowed $CV_T$ ($p = 0.001$) (Fig. 3a).

*Effects of verapamil on cardiac physiology.* In contrast to the multiple effects of 4-AP, the effects of verapamil, an L-type calcium channel blocker ($I_{CaL}$) was specific to the upstroke of the calcium transient (Fig. 3a). Verapamil prolonged $Ca^{2+}$ RT at all pacing rates tested as shown in Fig. 3b ($p < 0.001$). This could

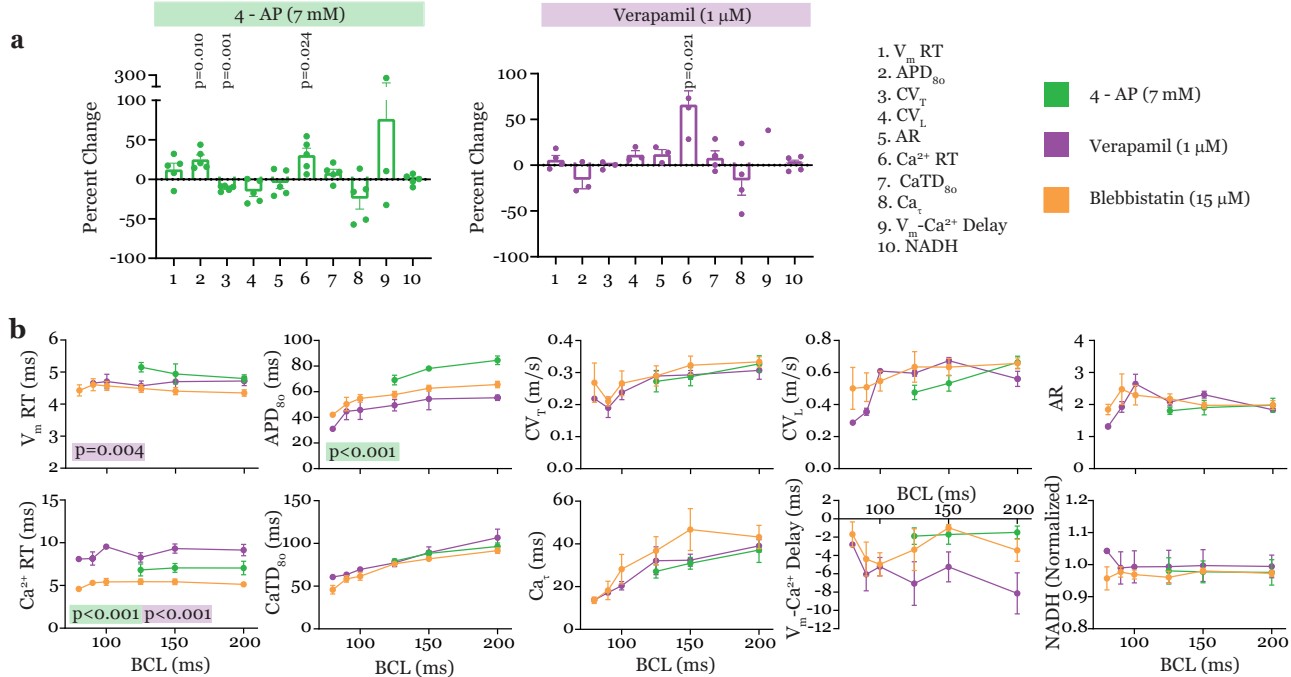

**Fig. 3 Triple-parametric optical mapping in drug testing. a** Ten-parameter panels demonstrating effects of 4-AP (7 mM, left, green) and Verapamil (1 μM, right, purple). **b** Summary restitution properties of the ten parameters measured simultaneously by triple-parametric optical mapping. Data presented as mean ± s.e.m., $n = 5$ hearts. Least squares regression analysis was performed for data in panel (**b**) to detect significant differences in parameters during Blebbistatin versus drugs (4-AP, Verapamil) treatment and all significant $p$ values are reported. Two-tailed, paired $t$-tests were performed for data in panel (**a**) to detect significant change induced by drugs compared to Blebbistatin and all significant $p$ values are reported. Benjamini–Hochberg correction was applied to account for multiple comparisons.

have contributed to the increase in $V_m$ RT by this drug ($p = 0.004$).

**Acute modulation of cardiac physiology by no-flow ischemia.** A separate set of hearts was perfused with a Control solution and a short episode of ischemia was induced by turning off the perfusion to the heart for a 5 min period followed by reperfusion. Ischemia modulated multiple parameters of cardiac physiology as shown in Fig. 4 while reperfusion restored all of them to pre-ischemic (baseline) values. Activation/intensity maps during baseline, ischemia (5 min), and reperfusion (5 min) are shown in Fig. 4a. All activation/intensity maps are from the same heart and all three maps in each column were generated from optical data that were simultaneously recorded. Time-dependent responses of the seven tested parameters during ischemia and reperfusion are illustrated in the graphs in Fig. 4b and Supplementary Data 2 while TPP graphs for ischemia and reperfusion demonstrate significant modulation of cardiac physiology during ischemia and restoration during reperfusion are shown in Fig. 4c. Significant modulation of each parameter in each heart was determined with respect to the baseline (pre-ischemic, $t = 0$) value from that same heart. Since no changes in any of the measured parameters are expected over the short duration (10 min) of this protocol without any external perturbations, baseline values serve as a control.

As expected, the first parameter that was significantly altered during ischemia was NADH intensity. A quick increase in NADH levels was observed, as early as 1 min into ischemia ($p = 0.024$). This was followed by changes in electrophysiology. Specifically, ischemia slowed $CV_T$ at 3 min ($p = 0.003$) and then prolonged $V_m$ RT ($p = 0.010$) at 5 min. Changes in AR and $Ca^{2+}$ RT were not statistically significant during the 5 min ischemic protocol. Lastly, $V_m$-$Ca^{2+}$ delay was also decreased by

ischemia ($p = 0.016$), possibly due to prolonged $V_m$ upstroke but unaffected $Ca^{2+}$ upstroke.

Reperfusion restored all tested parameters to pre-ischemic baseline values as quickly as 1 min after the start of perfusion. Such a quick response is possible due to the short duration of the preceding ischemia.

## Discussion

We present here the first application of triple-parametric optical mapping for simultaneous measurements of $V_m$, $Ca^{2+}$, and NADH, which allows studying MECC. We demonstrated the significance of this methodology in drug testing and cardiac disease studies by performing triple-parametric optical mapping in mouse hearts during blebbistatin, 4-AP, and verapamil treatments as well as during ischemia and reperfusion. We report that while blebbistatin and 4-AP modulated multiple parameters of cardiac physiology, the effects of verapamil were focused to a single parameter. Specifically, verapamil treatment-induced prolongation of $Ca^{2+}$ RT could be expected with an $I_{CaL}$ blocker. On the other hand, blebbistatin prolonged $V_m$ and $Ca^{2+}$ RTs while 4-AP caused prolongation of APD, $Ca^{2+}$ RT as well as slowing of $CV_T$. This methodology was also applied to investigate the acute effects of ischemia and reperfusion. While ischemia affected multiple parameters including increase in $V_m$ RT and NADH as well a decrease in $CV_T$ and $V_m$-$Ca^{2+}$ delay, reperfusion restored all these parameters to baseline values. By simultaneously measuring multiple aspects of cardiac function, we determined that changes in the metabolic state precede the electrophysiological modulation during ischemia. Thus, by implementing triple-parametric optical mapping, we determined unexpected off-target effects of drugs and sequence of modulation of cardiac physiology in disease.

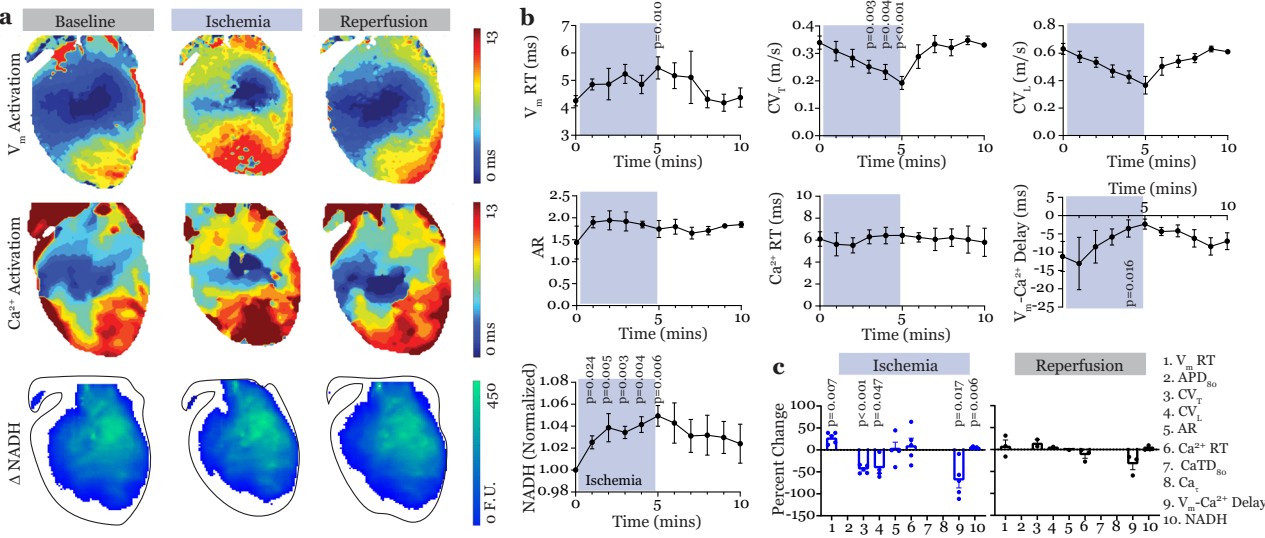

**Fig. 4 Triple-parametric optical mapping in cardiac disease assessment. a** Representative $V_m$ activation maps (top), $Ca^{2+}$ activation maps (middle), and NADH intensity maps (bottom) were recorded from the same mouse heart during baseline, ischemia, and reperfusion conditions. All maps in a given column were obtained by simultaneous imaging. **b** Summary of seven parameters measured over time, simultaneously by triple-parametric optical mapping. Three parameters—$APD_{80}$, $CaTD_{80}$, and $Ca^{2+}$ decay were not measurable due to motion artifacts in the repolarization phase due to absence of electromechanical uncoupling. The values reported are normalized to baseline ($t = 0$) for each heart in order to determine modulation of each parameter in each heart with respect to its own baseline condition. **c** Ten-parameter panels (only seven parameters reported) demonstrating changes in cardiac physiology during ischemia and restoration during reperfusion. Data are presented as mean ± s.e.m., $n = 5$ hearts. Two-tailed, paired $t$-tests were performed to detect significant change induced by ischemia and reperfusion compared to baseline and all significant $p$ values are reported. Benjamini–Hochberg correction was applied to account for multiple comparisons.

Blebbistatin is a selective inhibitor of myosin II isoforms found in skeletal muscles with little to no effect on other myosin isoforms. blebbistatin binds to the myosin-ADP-Pi complex and interferes with the phosphate release process, leaving the myosin detached from actin thereby arresting cellular contraction and preventing energy consumption by contraction thus, reducing metabolic demand[17]. Blebbistatin is widely used as an electromechanical uncoupler to study cardiac physiology by optical methods which require arresting the heart to prevent motion-induced artifacts. Blebbistatin is more advantageous to previously used electromechanical uncouplers in that its effects on cardiac physiology are minimal[18,19]. Blebbistatin does not alter calcium transient amplitude, rise time or decay as well as effective refractory period and ECG parameters[18]. However, mixed reports on its effects on rabbit APD have been previously published with groups demonstrating that blebbistatin either does not alter APD or that it prolongs APD[18,20,21]. Differences in methodologies including experimental conditions, poor perfusion, and motion correction algorithms applied could account for some of these differences in results. For example, blebbistatin applied to an ischemic preparation is likely to reverse ischemia-induced APD shortening, appearing to prolong APD.

Although the effects of blebbistatin are well studied, these studies were mostly performed on rat or rabbit hearts and tested at a concentration range of 0.1–10 μM which is less than recently reported concentrations used in mouse hearts[22]. In this study, we used 15 μM blebbistatin to arrest heart motion during optical mapping and determined the effects of this concentration of blebbistatin on mouse cardiac physiology. Blebbistatin altered two of the ten parameters measured in this study at a "normal" pacing rate (150 ms BCL, 400 bpm) for ex vivo hearts. The upstroke rise time of action potentials and calcium transients were prolonged during blebbistatin treatment. Additionally, at a slower heart rate (200 ms BCL), $CV_T$ was faster and this correlated with reduced NADH autofluorescence intensity. Lower NADH values could

correspond to increased ATP availability which could, in turn, modulate the ion channel and gap junction activity that affect cardiac conduction. This example illustrates how the use of triple-parametric optical mapping can uncover such interrelated MECC that support cardiac physiology.

Blebbistatin is fluorescent with solvent-specific spectral properties which could interfere with the measurement of relative changes in NADH autofluorescence intensity between treatments. Blebbistatin dissolved in DMSO (solvent used in this study) has an excitation/emission peak of 420/560 nm and the majority of the emission is above 500 nm[23]. The design of this triple-parametric optical mapping system filters NADH optical signals using a 450 ± 50 nm filter, thus avoiding the addition of blebbistatin fluorescence in NADH signals. Furthermore, exposure of blebbistatin-perfused hearts to UV light, at intensities and durations required for NADH imaging, does not cause cytotoxicity or significant changes in its electromechanical uncoupling properties[23].

4-AP is a potent inhibitor of the $I_{to}$ currents and is used in the treatment of multiple sclerosis. Inhibition of $I_{to}$, a Phase 1 repolarizing current could cause prolongation of APD. In cardiac tissue, 4-AP has been demonstrated to have a biphasic effect where APD shortening[24] is observed at lower concentrations but APD prolongation is induced at higher concentrations (~>5 mM)[25–27]. This could be due to the inhibitory effects of 4-AP on other ion currents like $I_{Kur}$ and hERG[26,28]. In the presence of isoproterenol, 4-AP also promotes EADs and DADs in cardiomyocytes[29].

In this study, we observed APD prolongation by 7 mM 4-AP treatment, as expected, at all studied pacing rates. Additionally, this APD prolongation prevented 1:1 capture at pacing at rates faster than 125 ms BCL in mouse hearts. However, we report here that the effects of 4-AP on cardiac physiology extend beyond the expected blocking of the $I_{to}$ current.

An inverse relationship between Phase 1 repolarization and calcium transient amplitude has been reported[30]. Decrease in

Phase 1 repolarization rate has been demonstrated to increase $I_{CaL}$, calcium transient amplitude and rise time. In our study, we also report an increase in the calcium transient rise time in hearts treated with 4-AP further supporting this relationship. However, the non-ratiometric calcium dyes used in this study did not allow the accurate quantification of calcium transient amplitude.

Lastly, we also demonstrated that 4-AP slows $CV_T$ in mouse hearts. Although the effects of 4-AP on cardiac conduction has not been previously reported, it has been demonstrated to restore conduction in injured neurons[31,32]. Although we did not test the effects of 4-AP in the context of injury, we report that 7 mM 4-AP reduces conduction velocity in mouse hearts. The dose dependence and underlying mechanism of this response will need further investigation.

Verapamil, an $I_{CaL}$ blocker used to treat angina, hypertension, tachycardia and other cardiac diseases also has hERG channel blocking properties[33,34]. It is probably due to its inhibitory effect on both potassium and calcium currents that the APD response to verapamil treatment has produced mixed results in previous studies. Verapamil-induced APD prolongation, shortening and no change have been previously reported[35–37]. The varying dose-dependent effects of verapamil on $I_{CaL}$ versus hERG current as well as differences in experimental models and tissues could explain some of these differing results. Verapamil has also been reported to have age-dependent effects on cardiac electrophysiology[38]. In this study, we report no statistically significant changes in APD in mouse hearts treated with 1 μM verapamil.

Furthermore, verapamil does not alter depolarization-related parameters. Verapamil has been reported to not alter the rate of depolarization in cells with sodium-dependent depolarization[39] or alter conduction velocity[40]. In line with these findings, we report no statistically significant changes in $V_m$ RT and CV in mouse hearts treated with verapamil.

The effects of verapamil on calcium handling are many. Inhibition of $I_{CaL}$ by verapamil has been shown to reduce the amplitude of calcium transients and contractility[35,41]. It has also been demonstrated to suppress calcium transient alternans and reduce spontaneous calcium release[42,43]. We report here prolonged $Ca^{2+}$ RT in verapamil-treated mouse hearts. An increase in $Ca^{2+}$ RT despite reduced calcium transient amplitude could suggest significantly decreased $I_{CaL}$ and calcium release from the sarcoplasmic reticulum. Lastly, we also report here that verapamil did not induce any significant changes in calcium reuptake as indicated by no changes in CaTD and $Ca_\tau$ parameters.

Ischemia is a condition, which is caused by the reduction or lack of blood supply to heart tissue. Ischemia modulates multiple parameters of cardiac physiology, including all three components of physiology measured in this study. Although the acute and chronic effects of ischemia are well-established, this study is the first to simultaneously assess cardiac electrical, calcium handling and metabolic functions to determine the complex sequence of MECC. Some of the well-known effects of acute ischemia include ATP reduction (NADH increase), APD shortening, $V_m$ RT increase, CV slowing, calcium alternans, and spontaneous calcium release[6,13,14,16].

Changes in NADH due to acute cardiac ischemia occur within 15 s and reperfusion can restore it to baseline within 60 s[44]. In this study, we used a 5 min no-flow ischemia model to measure the changes in cardiac physiology during acute ischemia and reperfusion. Ischemia increased NADH levels in the tissue at the earliest time point measured (1 min) and remained elevated throughout the ischemic period. This was the first of the ten parameters measured to be modulated suggesting that ATP depletion underlies most other physiological effects of ischemia. Next, at 3 min of ischemia, $CV_T$ slowing was observed. This was

followed an increase in $V_m$ RT. The effect of reduced ATP on the phosphorylation state of depolarizing sodium current and gap junctions could underlie these effects. It is also important to note that the effects of ischemia on $V_m$ RT could be underestimated because the optical action potential recorded in each pixel is an average of multiple cardiomyocytes. Therefore, it is possible that $V_m$ RT is increased sooner in the ischemic period than measured with this approach. Lastly, ischemia also reduced the $V_m$-$Ca^{2+}$ delay possibly due to slower depolarization (increased $V_m$ RT).

Reperfusion restored all measured parameters to pre-ischemic values within 1 min of restarting the perfusion to the heart. The short duration of the ischemic period could account for the immediate return to baseline conditions. Future studies aimed at determining the sequence of restoration of cardiac physiology during reperfusion could include prolonged ischemia periods or more frequent recordings during the reperfusion period.

In conclusion, this study demonstrates for the first time the application of triple-parametric optical mapping, which allows studying metabolism-excitation-contraction coupling in the heart. Here, we applied this methodology for drug cardiotoxicity testing and to study the modulation of cardiac physiology during ischemia/ reperfusion. We identified ten parameters of cardiac physiology related to electrical excitation, calcium handling and metabolism that give important information on the state of the heart. We developed a ten-parameter panel (TPP) graph which can give a quick overview of the effects of drugs or diseases on the heart. Using this approach, we determined the effects of blebbistatin, 4-AP, and verapamil on mouse cardiac physiology. While blebbistatin and 4-AP altered multiple aspects of cardiac physiology, the effects of verapamil were limited to calcium transient upstroke as expected with a calcium channel blocker. This demonstrates that triple-parametric optical mapping is a valuable tool to study cardiotoxicity of drugs in pre-clinical trials, particularly to identify off-target effects. Current drug testing is limited primarily to QT interval testing. This field could greatly benefit from a more comprehensive assessment of cardiac physiology as is the case with triple-parametric optical mapping. Lastly, we also applied this methodology to determine the sequence of modulation of the multiple facets of cardiac physiology during acute ischemia. Simultaneously measuring the three facets of cardiac physiology identified that changes in metabolism during acute ischemia precede the effects on electrophysiology. The critical applications of this methodology demonstrate the need and the significance of triple-parametric optical mapping.

Limitations of the study include the following. The effects of ischemia on repolarization/calcium reuptake parameters were not measurable because optical mapping was only performed in the absence of the electromechanical uncoupler blebbistatin. This approach was implemented because the changes in NADH due to ischemia were not accurate in the electromechanically uncoupled hearts, as blebbistatin reduces metabolic demand. Another limitation of blebbistatin is its fluorescence overlapping with NADH, which requires careful consideration of the data recorded from blebbistatin-treated hearts in metabolically compromised states. Hearts were only subjected to 5 min of ischemia because beyond this time point, the quality of the optical signals deteriorated and did not allow appropriate analysis. Future studies will explore newer dyes with improved signal quality even during ischemia.

## Methods

All experimental protocols were approved by the Institutional Animal Care and Use Committee at George Washington University and are in accordance with the National Institutes of Health Guide for the Care and Use of Laboratory Animals.

**Triple-parametric optical mapping system set up and alignment**. The triple-parametric optical mapping system uses three CMOS cameras (MiCAM05, Sci-Media) and the MiCAM05 digital interface for 3–4 camera (SciMedia). BV workbench 2.6.1 (SciMedia) was the software used for camera alignment and

acquisition. This system triggers all attached cameras simultaneously. The cameras have $100 \times 100$ pixel resolution with a camera sensor of $10\ mm \times 10\ mm$ dimension. A sampling time of 1 ms (1 kHz frequency) was used to acquire the data file of 2 s duration in this study.

All cameras are focused on the same field of view through a tandem lens configuration as illustrated in Fig. 1a. All filter cubes, mechanical and stage components of the system were custom 3D printed and system assembly instructions were previously published for dual parameter optical mapping[45]. The camera cages were specifically designed for SciMedia cameras but the other system components can be used with optical mapping systems from other sources. Two modifications were made to these parts to accommodate an extra camera in this current system (Fig. 1b). Open-source files of these components are also available at Github (https://github.com/optocardiography).

Two light-emitting diodes (LED) excitation light sources at 365 nm (Mightex Systems, LCS-0365-04-22) and $520 \pm 5$ nm (Prizmatix, UHP-Mic-LED-520) with collimators were used in episcopic illumination mode to excite the dyes or induce autofluorescence in the cardiac tissue. The emitted light is collected by an infinity-corrected, planapo 1X lens (objective lens, SciMedia) with a working distance of 61.5 mm. This lens then focuses on the collected light at infinity. This infinity correction allows multiple cameras to be introduced in the light path. Light at different wavelengths were separated using dichroic mirrors and filters as illustrated in Fig. 1c and passed through a second planapo 1X lens (projection lens, SciMedia) before being recorded using the CMOS cameras. The use of two lenses of the same focal length in tandem lens configuration results in a magnification of 1X (pixel size = 0.1 mm)[46]. First, light below 510 nm is split from the straight light path by a dichroic mirror which is then filtered by a $450 \pm 50$ nm filter and directed to the first CMOS camera to record NADH autofluorescence. Next, Rhod2-AM (intracellular calcium) signal is split from the straight light path using a 630 nm dichroic mirror, filtered using a $590 \pm 33$ nm filter, and recorded using the second CMOS camera. Finally, the RH237 (transmembrane potential) signals at wavelengths above 630 nm are filtered through a $690 \pm 50$ nm filter and recorded using the third CMOS camera. With this optical system, the amount of emission spectral overlap between NADH and Rhod2-AM as well as between Rhod2-AM and RH237 can be reduced and the interference from a different channel is minimized.

At the start of each experiment, all three cameras were focused and aligned. Each camera was attached to its own projection lens and focused at infinity. The projection lens/camera unit was then attached to the filter cubes. A focusing target was positioned in front of the objective lens and the position of the target was adjusted until all three cameras were in focus. Next, the three cameras were spatially aligned using the Camera Calibration function in the BV Workbench software (Brainvision). This feature overlays the image captured by the different cameras and uses an edge detection algorithm to allow the user to manually adjust the angle of the dichroic mirrors until all fields of view are spatially aligned. Once, all cameras were focused and aligned the system is ready for use.

**Langendorff perfusion**. Adult male and female mice on a C57BL/6 background were used in this study. Mice were anesthetized by isoflurane inhalation and cervical dislocation was performed. Hearts were quickly excised following thoracotomy and the aorta was cannulated. The heart was attached to a Langendorff perfusion system, hung in a vertical position in a temperature-controlled bath and perfused with a modified Tyrode's solution containing (in mM) 130 NaCl, 24 $NaHCO_3$, 1.2 $NaH_2PO_4$, 1 $MgCl_2$, 5.6 Glucose, 4 KCl, 1.8 $CaCl_2$, pH 7.40 and bubbled with carbogen (95% $O_2$ and 5% $CO_2$) at 37 °C. Perfusion pressure was maintained at ~80 mmHg by adjusting the flow rate between 1 and 1.5 ml/min. A platinum bipolar electrode was placed at the center of the anterior surface residing in the middle of the field of view. Gentle pressure was applied to the back of the heart as it was pushed up to the front optical glass of the bath using a paddle, allowing the pacing electrode to be held in place. Electrical stimuli were applied to determine the threshold of pacing. Hearts were paced at 1.5X pacing threshold amplitude and 2 ms stimulus duration, only during optical recording. Hearts were paced at varying rates as listed in the figures to determine the restitution properties of the heart under each condition.

**Optical mapping**. The heart was subjected to a 10 min equilibration period followed by staining with the voltage- and calcium-sensitive dyes. A 1 ml mixture of RH237 (30 μl of 1.25 mg/ml dye stock solution + 970 μl Tyrode's solution, Biotium 61018) was prepared and immediately injected into the dye port above the cannula over a 3–5 min period. This was followed by a 5 min washout period. Similarly, a 1 ml mixture of Rhod2-AM (30 ul of 1 mg/ml dye stock solution + 30 μl Pluronic F-127 + 940 μl Tyrode's solution, Thermo Fisher Scientific R1244 and Biotium 59005, respectively) was prepared and immediately injected into the dye port over a 3–5 min period. This was followed by 5 min dye washout period.

The heart was then illuminated by two LED excitation light sources at 365 nm and $520 \pm 5$ nm wavelengths. While the former induces autofluorescence of NADH in the tissue, the latter excites both RH237 and Rhod2-AM dyes. Control optical recordings of NADH, $V_m$, and $Ca^{2+}$ were simultaneously acquired at 1 kHz sampling rate.

**Electromechanical uncoupling and drug treatment**. After Control recordings, hearts were treated with 15 μM blebbistatin (Cayman Chemicals 13186) for 20 min and optical recordings were acquired as above. Next, hearts were treated with 4-AP (7 mM, Millipore Sigma 275875) and verapamil (1 μM, Sigma Aldrich V4629) one at a time, in the presence of blebbistatin. Optical recordings were once again acquired for each of these conditions as described above.

**Ischemia**. In a separate set of hearts, after the initial equilibration period and dye staining/washout, hearts were subjected to 5 min of no-flow ischemia followed by 5 min of reperfusion. Hearts were continually paced at 300 ms BCL throughout this period. Optical recordings were acquired prior to (baseline), during (ischemia), and after (reperfusion) this period of no perfusion at 1 min intervals. Ischemia period was limited to 5 min because beyond this point, the optical signals were of very poor quality which would not allow suitable analysis. Signal quality deteriorates during periods of ischemia due to poor or no perfusion of the tissue and quickly recovers during reperfusion. Poor quality signals are defined as those with low signal-to-noise ratios that would not allow for the accurate measurements of the parameters in this study.

**Data analysis**. Optical data of NADH, transmembrane potential and calcium were analyzed using a custom Matlab software, Rhythm 3.0, which is available in an open-source format on Github. Rhythm 3.0 which is an upgraded version of Rhythm 1.2, incorporates NADH visualization and analysis features as well as the ability to calculate delay in transmembrane potential to intracellular calcium activation ($V_m$-$Ca^{2+}$ Delay). Rhythm software was written to analyze data formats generated by SciMedia systems but can be modified to analyze other data formats. In this study, ten different parameters were measured from the three optical signals that were simultaneously recorded under each condition. These include, from transmembrane potential: (1) rise time ($V_m$ RT), (2) action potential duration ($APD_{80}$), (3) transverse and (4) longitudinal conduction velocity ($CV_T$ and $CV_L$), (5) anisotropic ratio (AR), from intracellular calcium: (6) rise time ($Ca^{2+}$ RT), (7) calcium transient duration ($CaTD_{80}$), (8) calcium decay time constant (τ), (9) $V_m$-$Ca^{2+}$ delay, and (10) NADH fluorescence intensity. The rate-dependence of each of these parameters is indicated in the parameter vs BCL graphs while the summary of effects of each condition is illustrated in the TPP graphs. The values reported in the TPP graphs correspond to BCL = 150 ms which is considered "normal" heart rate (400 bpm) for ex vivo mouse heart preparations.

Upstroke rise time (RT) from $V_m$ and $Ca^{2+}$ signals were measured as the time period from 20 to 90% of the upstroke of the action potential and calcium transient, respectively. $APD_{80}$ and $CaTD_{80}$ were measured as the time interval between activation time (time of maximum first derivative of the upstroke) and 80% of repolarization and calcium transient decay, respectively. $CV_L$ and $CV_T$ in the parallel and perpendicular direction to fiber orientation, respectively, were calculated using differences in activation times and known interpixel distances. The anisotropic ratio was calculated as the ratio of $CV_L$ to $CV_T$. Calcium decay time constant (τ) was determined by fitting an exponential to the last 50% (50–100%) of the calcium transient decay phase. $V_m$-$Ca^{2+}$ delay was defined as the time interval between the activation time of the $V_m$ signal minus the activation time of the $Ca^{2+}$ signal. Lastly, NADH intensity was measured as the average of the absolute autofluorescence intensity value in the optical recording. NADH intensity in each heart was normalized to the first measured NADH intensity value from that particular heart in order to avoid interexperimental variability.

**Statistics and reproducibility**. All data are reported as mean ± standard error of the mean (SEM). A sample size of five hearts was used for all groups, drug treatment and ischemia protocols. An alpha level of 0.05 was used in all tests.

Regression analysis was performed on the restitution data using Graphpad Prism Version 9.3.1 software. Nonlinear regression analysis was performed using the least squares regression fitting method. For parameters that exhibited restitution property ($CV_T$, $CV_L$, AR, $APD_{80}$, $CaTD_{80}$, and Ca τ), an exponential plateau model ($Y = Y_M - (Y_M - Y_0)e^{-kX}$) was used. For Vm-Ca delay parameter, the third order polynomial model ($Y = B_0 + B_1X + B_2X^2 + B_3X^3$) was used. For all others a simple linear regression model ($Y = a + bX$) was used. All other parameters were left at default settings in this software. Good model fit was determined by r-squared value >0.5 (except Vm-Ca delay during blebbistatin treatment) and by confirming the random nature of the residual plots.

The parameters of the best fit were compared between groups (Control vs Blebbistatin, Blebbistatin vs 4-AP, and Blebbistatin vs Verapamil) using the extra sum-of-squares F test. Two-tailed paired t-tests were performed for all other data. Since multiple statistical tests were performed on this data set (ten parameters, five treatments), Benjamini–Hochberg correction was applied with a false discovery rate of 20%. Shapiro–Wilk test was applied to test for data normality of the raw data values as well as the best-fit parameters of the regression analysis and >95% data sets passed the normality test. However, the small sample size of this data set may be a limitation.

For the summary data in Figs. 2d, 3a, and 4, paired two-tailed t-tests were performed. In the case of Figs. 2d, 3a, and 4c, percent change in each parameter by a given drug (Blebbistatin, 4-AP, and Verapamil) versus Control was compared. In Fig. 4b, statistical tests compared change in each parameter at a given time point

versus Baseline ($t = 0$). Benjamini– Hochberg correction was applied with a false discovery rate of 20% to account for multiple comparisons.

**Reporting summary**. Further information on research design is available in the Nature Research Reporting Summary linked to this article.

## Data availability
All data will be available upon request to the corresponding authors.

## Code availability
Rhythm 3.0 is available at Github (github.com/optocardiography) in an open-source format. DOI: 10.5281/zenodo.5784023.

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

## Acknowledgements
This work was funded by Leducq Foundation (project RHYTHM), American Heart Association SCD SFRN grant, and NIH grant R44 HL139248 to I.R.E. and an American Heart Association Postdoctoral Fellowship (19POST34370122) to S.A.G.

## Author contributions

Study conception: S.A.G. and I.R.E.; Study design: S.A.G.; Data acquisition, analysis, and interpretation: S.A.G.; Designing and 3D printing hardware: S.A.G. and Z.L.; Preparing analysis software: S.A.G.; Manuscript drafting and figure preparation: S.A.G.; Revision and final version approval: S.A.G., Z.L., and I.R.E.

## Competing interests

The authors declare no competing interests.
