## [Peer Review File · Communications Biology]

Reviewers' comments:

Reviewer #1 (Remarks to the Author):

1. The manuscript submitted presents a custom built, simple optical setup capable of fluorescence imaging of cardiac tissue with three different channels that operate simultaneously. The article is presented quite clearly, although it is oriented at an audience with a clear idea of cardiac physiology, and with a solid knowledge of medical terminology. The authors nevertheless give enough background in the first paragraph of the Introduction to ensure a better understanding of the problem at hand. However, a reference to a review article or a textbook should be added, not as a validation of their explanation, but to point to a resource for further documentation.
2. A significant drawback in the understanding of the Results section is that the numerous abbreviations are systematically introduced only in the Online Methods, which in the manuscript I reviewed were reported at the end. This further creates confusion in the numbering of the figures, since Figure 1 is not referenced until then, and is not mentioned until the last paragraphs. It would thus be beneficial to give a small introduction to the optical system before the "Blebbistatin modulates cardiac physiology:" paragraph, or as a separate Supplementary material, or with other solutions compatible with editorial policies and author's choice. While this issue is of course not fully due to the authors, it should be kept in mind for further manuscript versions.
3. Given the large number of parameters the authors employ in their analysis, a visual explanation of the meaning of the parameters is recommended and would make the message clearer, much as the diagram in Figure 1C helps the reader understand the choice of fluorophores by the authors. This is especially crucial given the choice of the authors to employ blebbistatin to get rid of motion artifacts due to cardiac contractions, and thus would further motivate their choice.
4. The long sentence in lines 117-120 is made up of two sentences joined by an "and" conjunction, but seem unrelated. I would suggest splitting them. In the same paragraph, it is not clear why three of the parameters are not calculated in the control sample. Is it due to the motion artifacts previously mentioned? In any case, a clear motivation for this should be stated.
5. In Figure 2b, the authors mention the observation of the heart during Control and Blebbistatin treatment. However, the images look very different, thereby raising the doubt whether it is the same sample or a different one. This should be made clear in the text.
6. Concerning the statistical comparison, some concerns arise by considering the graphs provided in Figure 2. First of all, the authors mention the Bonferroni correction, but report no reference for it. Furthermore, it is not clear to me how this applies in this circumstance. Given Figure 2c, and what emerges from the text, it appears that the authors are performing measurements of their different metrics at different BCL values, and comparing what they obtain between control group and blebbistatin group, at each BCL. This does not appear to the reviewer a case in which Bonferroni correction is relevant, since it involves just the comparison between two groups for each individual metric, at each individual value of BCL. Therefore, it is also not clear which number is used to adjust the significance value through the Bonferroni correction, and what is the rationale for adopting it. Instead, this would apply to Figure 2d, assuming that the authors mean to use each value of the BCL as a different group. The authors would need to state this clearly in the text as this is crucial for understanding their methodology. The same comment would apply also to Figure 3, although in this case the correction is probably also used at each BCL for each metric, given the presence of two treatments. However, the * notation is too simplistic to help the reader distinguish which groups are significantly different. In any case, the "Statistics" section of the online methods clearly deserves significant expansion and careful writing, since its centrality in the message the authors want to deliver.
7. In Figure 4b, the absence of the values for the control group in these graphs make the statistical comparison and the reported significance levels quite obscure, and it would be beneficial to add them.
8. As a general comment, NADH fluorescence is always reported as "normalized", however the reference value of the normalization is not clear, since the meaning of "control" arguably changes in each of the performed experiments.
9. The acronym BCL is never explained.
10. Concerning the Online Methods sections, while I appreciate the effort to deliver a detailed GitHub repository for the data analysis code and optical setup, I would personally suggest to adopt a different platform, such as Zenodo, so that it would have a DOI and a future reader could see the state of the methods at the time of writing, and linking the GitHub repository to give the

opportunity of following the latest developments. While an emphasis on simultaneous acquisition is given throughout the text, it is not clear from this section how this is implemented. How the cameras are triggered for acquisition, and the acquisition parameters adopted are left untold, and even a brief mention is necessary to understand how the authors operated. While I understand the care that the authors put into compiling the optical component list, it is still necessary to use proper terminology. The kind of excitation light source, the presence of condensers to illuminate the sample, are simply not reported. Furthermore, why the authors choose to disclose some details in the text, such as the cutoff wavelengths of the dichroic mirrors or the operating range of the bandpass filters, but withhold some (such as the focal lengths of the lenses in front of the cameras and the resulting effective magnification or pixel size) is unclear. Finally, while all the designs are freely released in an "open source" way (despite no license information being present in the GitHub repository), the necessary montage instructions are not provided. How the software mentioned helps the spatial alignment of the cameras is also not clear. More details on the treatment protocols of the heart samples would also be beneficial. For example, it has been very difficult for the present reviewer to retrieve spectral information concerning the RH237 dye, without knowing the supplier.

11. In line 403, the "very poor quality" remark is somewhat mysterious. The authors should elaborate on what does this mean. Is it due to phototoxicity, fluorescence bleaching, or some other factor?

12. Version 3.0 of the Rhythm software is not online, please rectify.

13. In line 428, the authors should state clearly NADH fluorescence, instead of just NADH

14. The "Statistics" section is definitely incomplete and not sufficiently elaborated, as already stated above. Please expand with all the necessary details to replicate the statistical analysis performed by either the authors or their software.

Reviewer #2 (Remarks to the Author):

This is a very nice methods paper, with good rationale and results. It is well written and code-sharing (for analysis and 3D printing design) is commendable. Also it shows a path towards more complex assessment of drug effects, which is important, as current assessment techniques do not provide a complete enough picture. I also like that the techniques described do not appear to be massively complex, so this is something that can be easily picked up by other labs.

Questions/comments:

1) The spectral overlap between RH-237 and Rhod-2 is unpleasant, but I believe most optical mapping labs using the dye combination found a way of working with it. However, the emission spectrum of the NADH autofluorescence makes me wonder about the overlap with Rhod-2. Could that affect features like diastolic Ca assessment? (I know that Rhod-2 is not ratiometric, but e.g. in ischemia one can compare fluorescence baseline before/after nevertheless). Would it make sense to set the illumination sources in a way that a part of the field of view would report V and Ca signals, with the other part reporting NADH autofluorescence? (these then obviously wouldn't be colocalized, but I'm not sure that's a critical limitation). Or could a focused beam be used to evoke NADH autofluorescence in a small part of the field of view, allowing the rest of the FOV to not suffer from the possible spectral overlap?

2) Minor question – was the solution in the Langendorff prep recycled? (I find the continuous perfusion with diluted dyes via recycling improves signal quality and dye longevity)

3) The statistical analysis contains aspects that are not ideal. Despite my questions/comments, I do appreciate that the concrete results are not the central point of this paper, so I do not think it is critical what exactly is significant in this paper.

a. Why was ANOVA used? I did not see the p-values reported or commented. One problem is that ANOVA is highly sensitive to the assumption of equal variance (here between different bIcs) – and while I cannot be absolutely sure without seeing the raw data, this looks almost certainly violated (e.g. 2c, CV_T or V-Ca delay). The second issue is that ANOVA treats distinct bIcs as independent – which they are not – they are ordered. ANOVA will give the same p-value for different reordering of your x-axis, while the biological interpretations of such data would be often different.

Perhaps the most clear example is dose-response experiments, which is another example of ordered x-axis where people sometimes use ANOVA. If, for four increasing doses, the feature of interest takes mean values such as 0, 1, 2, 3 (with some sensible variance), it can be read as a straightforward dose-dependent effect. However, if the means are ordered as 2, 1, 3, 0, it looks rather like artefact or some very nontrivial dose-dependence. Yet ANOVA would give you the same p-value for these two scenarios.

For this reason, it is often recommended to do regression analysis instead. Your data are often relatively linear, so linear regression could be used (or you could fit an exponential curve in the cases of restitution-like curves). E.g. in Fig 2c/NADH, you could take, for each heart, $x = \text{bcls}$, $y =$ the difference between blebbistatin and control at the bcl, fit a line to this – do this for all the hearts – and then do a t-test (or another test if assumptions not holding) on the offset or slope across the fitted lines (in 2c, most subplots would likely show a difference in offset, rather than in slope). You can find a bit more discussion of the ANOVA on dose-dependence (or bcl, which is similar case of ordered x-axis) here:

https://www.graphpad.com/guides/prism/latest/statistics/stat_qa_multiple_comparisons_after_.htm and here: https://www.graphpad.com/guides/prism/latest/statistics/stat_posttesttrend.htm

b. I don't think there is a problem just using paired t-tests with a correction (although mind that the same issue with ordering of the x axis being ignored persists – you are losing information by treating the bcls as independent). However, in this case, I would wonder about the assumption of normality – there are cases where the variance is quite uneven for a bcl at different conditions, so that might easily translate into non-normal distribution of the differences. It would be good to include supplementary plots of the distributions of the differences entering the t-tests (or just tables with data underlying the bcl-vs-feature plots so that curious readers can check themselves). I think visual inspection here is more useful than a normality test such as Shapiro-Wilk, which is highly underpowered on $n=5$, so even if it says the data are not significantly abnormal, it does not mean much.

c. A related point – please correct the wording in phrases such as “we report no change in APD in mouse hearts treated with 1 μM verapamil” – this is simply not true, there is a difference. Yes, it is often not significant, but there is almost always a difference. This is particularly important in a study with such low n 's, and possibly sometimes violated test assumptions – the fact that something was not significant does not mean much, and definitely not that there was “no difference”. See e.g., Arnheim et al. 2019 (<https://pubmed.ncbi.nlm.nih.gov/30894741/>) for how much pain and distress the phrase “there was no difference” causes to statisticians.

Here I'd just write something like “there was no statistically significant difference”/“verapamil did not induce a statistically significant changes in calcium reuptake” etc. – and clarify somewhere (e.g. Limitations) that the study has very low statistical power with $n=5$ and a large number of tests carried out with correction, so real effects were likely to be missed.

d. Bonferroni correction is very conservative and particularly on $n=5$, it is very likely to make you miss a lot of genuine effects. Did you consider using the Benjamini-Hochberg correction instead?

e. Why was s.e.m. used instead of standard deviation? Stdev is a generally preferable indicator of data spread, with s.e.m. often making the variability look smaller than it is. This is particularly critical with uneven group size, which is not a problem in your paper, but still, it does not give a great picture of the data variability. See e.g.

<https://diabetes.diabetesjournals.org/content/62/8/e15> or

<https://journals.plos.org/plosone/article?id=10.1371/journal.pone.0110364> . I appreciate that s.e.m. also has its place in statistical analysis, but in cases where you report it, it appears that a standard deviation would be more appropriate.

f. Your text is not always conforming to the statistical guidelines of the journal

<https://www.nature.com/commsbio/submit/submission-guidelines#statistical-guidelines> – e.g. accompanying significance statements with p-values.

Response to Reviewers:

We thank the reviewers and editors for the time and effort of reviewing our manuscript. Your critiques served us as a welcome guide during revisions and helped to significantly improve our manuscript. Below is a point-by-point discussion of how each of the reviewers' concerns were addressed. To facilitate the review process, we restated reviewers' comments in black font and provide our response in blue font.

Reviewer #1:

1. The manuscript submitted presents a custom built, simple optical setup capable of fluorescence imaging of cardiac tissue with three different channels that operate simultaneously. The article is presented quite clearly, although it is oriented at an audience with a clear idea of cardiac physiology, and with a solid knowledge of medical terminology. The authors nevertheless give enough background in the first paragraph of the Introduction to ensure a better understanding of the problem at hand. However, a reference to a review article or a textbook should be added, not as a validation of their explanation, but to point to a resource for further documentation.

We agree with the reviewer that providing further reading resources will be helpful to the readers. We have now included additional comprehensive reviews, both by our group and others. References 1 and 2 give an overview of basic concepts of cardiac electrophysiology and reference 3 and 4 are reviews on optical mapping.

2. A significant drawback in the understanding of the Results section is that the numerous abbreviations are systematically introduced only in the Online Methods, which in the manuscript I reviewed were reported at the end. This further creates confusion in the numbering of the figures, since Figure 1 is not referenced until then, and is not mentioned until the last paragraphs. It would thus be beneficial to give a small introduction to the optical system before the "Blebbistatin modulates cardiac physiology:" paragraph, or as a separate Supplementary material, or with other solutions compatible with editorial policies and author's choice. While this issue is of course not fully due to the authors, it should be kept in mind for further manuscript versions.

We apologize for confusion with abbreviations and figures! We thank the reviewer for bringing this to our attention. We have now included the paragraph in the beginning of the Results section to give an overview of the optical mapping system illustrated in Figure 1 as well as to introduce the abbreviations and define the parameters. This paragraph is on lines 100-135:

"Triple parametric optical mapping system was 3D printed and set up as illustrated in Figure 1a-b and the separation of signals of different wavelengths is illustrated in Figure 1c. All design files for 3D printed hardware (in STL format) and data analysis software (Matlab) are available under an open-source license at Github (<https://github.com/optocardiography>, DOI: 10.5281/zenodo.5784023). The following 10 parameters were measured from the optical recordings and an illustration of each parameter definition is included in Figure 1d. From the NADH recordings, the absolute intensity of the NADH signals was measured which corresponds to NADH concentration in the tissue. Since this parameter can vary between hearts depending on experimental conditions, all measurements from a given heart were normalized to the first

recording from that same heart (Control at 200 ms pacing rate for drug testing protocol and Baseline at 200 ms pacing rate for ischemia protocol). This allows the determination of changes in NADH induced by drug treatment or disease without confounding interexperimental variables. From the depolarization/calcium release phase of the V_m and Ca signals, V_m and Ca Rise Times (V_m RT and Ca RT), longitudinal and transverse conduction velocity (CV_L and CV_T), anisotropic ratio (AR), and activation delay between V_m and Ca traces (V_m -Ca delay) were calculated. Rise time was defined as the time taken for depolarization, from 20 to 90%. In the case of V_m RT, this parameter indicates function of depolarizing currents while in the case of Ca RT, this parameter indicates time taken for calcium entry into the cell and calcium-induced calcium release from the sarcoplasmic reticulum. Conduction velocity is the speed with which the activation wavefront travels in a given direction, longitudinal (parallel to fiber orientation) and transverse (perpendicular to fiber orientation). AR is the ratio of CV_L to CV_T and indicates ellipticity of the propagating wavefront. Higher AR is associated with increased arrhythmogenicity. V_m -Ca delay is calculated to determine the excitation-contraction coupling. Prolonged delay suggests uncoupling between electrical excitation and mechanical contraction. From the repolarization/calcium reuptake phase of the V_m and Ca signals, action potential duration at 80% repolarization (APD_{80}), calcium transient duration at 80% reuptake ($CaTD_{80}$), and calcium decay constant (Ca τ) were calculated. It is important to note that these 3 parameters (APD_{80} , $CaTD_{80}$ and Ca τ) were only measurable in hearts after Blebbistatin perfusion. Without Blebbistatin, motion artifacts were present in the signals which distort the V_m and Ca signals in the repolarization/reuptake phase (Figure 2a, top). APD_{80} was defined as the time interval between activation and repolarization or in other words the duration for one cardiac cycle. Similarly, $CaTD_{80}$ was defined as the time interval between calcium release and reuptake of calcium back into the sarcoplasmic reticulum. Both shortening and prolongation of these parameters have been reported to be arrhythmogenic. Lastly, Ca τ is the decay constant measured by fitting an exponential to the reuptake phase of the calcium transients. This parameter is indicative of how quickly calcium in the cytoplasm is removed after each contraction.”

3. Given the large number of parameters the authors employ in their analysis, a visual explanation of the meaning of the parameters is recommended and would make the message clearer, much as the diagram in Figure 1C helps the reader understand the choice of fluorophores by the authors. This is especially crucial given the choice of the authors to employ blebbistatin to get rid of motion artifacts due to cardiac contractions, and thus would further motivate their choice.

Thank you for this excellent suggestion! We agree that the definition of the many parameters typically used in the field can be confusing. As suggested, we have now included an illustration of each parameter definition in Figure 1d to demonstrate each parameter and how it was measured from the optical signals. The effects of Blebbistatin on reducing motion in optical signals are demonstrated in the next figure (Figure 2a).

4. The long sentence in lines 117-120 is made up of two sentences joined by an “and” conjunction, but seem unrelated. I would suggest splitting them. In the same paragraph, it is not clear why three of the parameters are not calculated in the control sample. Is it due to the motion artifacts previously mentioned? In any case, a clear motivation for this should be stated.

Thank you! This is very helpful recommendation. The long sentence has been shortened into two as suggested (new line 150-151). The reviewer is correct that the addition of Blebbistatin removes distortion of the repolarization phase of the optical signals and thereby allows the measurement of 3 additional parameters from the repolarization phase. This is further explained in lines 123-129 and 150-153.

“From the repolarization/calcium reuptake phase of the V_m and Ca signals, action potential duration at 80% repolarization (APD_{80}), calcium transient duration at 80% reuptake ($CaTD_{80}$), and calcium decay constant ($Ca \tau$) were calculated. It is important to note that these 3 parameters (APD_{80} , $CaTD_{80}$ and $Ca \tau$) were only measurable in hearts after Blebbistatin perfusion. Without Blebbistatin, motion artifacts were present in the signals which distort the V_m and Ca signals in the repolarization/reuptake phase (Figure 2a, top).”

“Treatment with blebbistatin (15 μM) abolished contractions and removed motion artifacts as shown in Figure 2a (bottom). Preventing motion-induced distortion of signals in the later phases of the action potential and calcium transient allowed the measurement of repolarization/calcium reuptake parameters such as APD_{80} , $CaTD_{80}$ and $Ca \tau$.”

5. In Figure 2b, the authors mention the observation of the heart during Control and Blebbistatin treatment. However, the images look very different, thereby raising the doubt whether it is the same sample or a different one. This should be made clear in the text.

Thank you for pointing out this apparent discrepancy. However, all activation/intensity maps shown in Figure 2b are from the same heart. The three upper maps are from simultaneous recordings from the heart during control conditions and the bottom maps are from simultaneous recordings after blebbistatin treatment. We added a clarification – see below.

If by the hearts being different, the reviewer is referring to the silhouette of the maps, it is because of the different signal intensities in the three optical channels due to the different sources of fluorescence (different dyes/autofluorescence). As part of signal conditioning, our custom Matlab program allows removing background noise from signals from the heart based on a defined threshold. Because of the different properties of the dyes and the NADH autofluorescence, some of the background pixels around the border of the heart will get picked up as signal which contributes to slightly different edges in the silhouette of the heart. However, these border pixels are not used in any of the analysis and is only present in the visual representation of the maps.

This point is further explained in lines 676-680.

“All maps were recorded from the same heart, the top three maps were from simultaneous recording during Control and the bottom three from simultaneous recording during Blebbistatin treatment. Slightly different silhouettes, particularly around the boundaries is due

to different background noise removal levels using a thresholding algorithm. These boundary pixels were not used in analysis.”

6. Concerning the statistical comparison, some concerns arise by considering the graphs provided in Figure 2. First of all, the authors mention the Bonferroni correction, but report no reference for it. Furthermore, it is not clear to me how this applies in this circumstance. Given Figure 2c, and what emerges from the text, it appears that the authors are performing measurements of their different metrics at different BCL values, and comparing what they obtain between control group and blebbistatin group, at each BCL. This does not appear to the reviewer a case in which Bonferroni correction is relevant, since it involves just the comparison between two groups for each individual metric, at each individual value of BCL. Therefore, it is also not clear which number is used to adjust the significance value through the Bonferroni correction, and what is the rationale for adopting it. Instead, this would apply to Figure 2d, assuming that the authors mean to use each value of the BCL as a different group. The authors would need to state this clearly in the text as this is crucial for understanding their methodology. The same comment would apply also to Figure 3, although in this case the correction is probably also used at each BCL for each metric, given the presence of two treatments. However, the * notation is too simplistic to help the reader distinguish which groups are significantly different. In any case, the “Statistics” section of the online methods clearly deserves significant expansion and careful writing, since its centrality in the message the authors want to deliver.

We thank the reviewer for bringing up this important issue. We have now performed new statistical analysis as recommended and have explained in detail all the statistical tests performed in the Statistics section (Lines 488-508). We have also explained the use of multiple comparison corrections. Additionally, we now describe in each figure legend the specific statistical test performed on each data set and the specific comparisons made.

“Statistics & Reproducibility: All data are reported as mean \pm standard error of the mean (SEM). A sample size of 5 hearts was used for all groups, drug treatment and ischemia protocols. An alpha level of 0.05 was used in all tests.

Least-squares regression analysis was performed on the restitution data, extra sum-of-squares F test for the best fit parameters from the regression analysis and two-tailed, paired t-tests for all other data. Since multiple statistical tests were performed on this data set (10 parameters, 5 treatments), Benjamini- Hochberg correction was applied with a false discovery rate of 20%. Shapiro-Wilk test was applied to test for data normality of the raw data values as well as the best-fit parameters of the regression analysis and >95% data sets passed the normality test. However, the small sample size of this data set may be a limitation.

For the restitution data (Figures 2 and 3), regression analysis was chosen due to the ordered nature of the independent variable (BCL). For parameters that exhibited restitution property (CV_T , CV_L , AR, APD_{80} , $CaTD_{80}$, Ca τ and V_m -Ca Delay), an exponential plateau model was used and for others a simple linear regression model was used. The parameters of the best fit were compared between groups (Control vs Blebbistatin, Blebbistatin vs 4-AP, Blebbistatin vs Verapamil) using the extra sum-of-squares F test.

For the summary data in Figure 2d, 3a and 4, paired two-tailed, t-tests were performed because it assumes normal distribution of data and does not assume equal variance. In the case of figures 2d, 3a and 4c, percent change in each parameter by a given drug (Blebbistatin, 4-AP and Verapamil) versus Control was compared. In Figure 4b, statistical tests compared change in each parameter at a given time point versus Baseline (t=0). ”

7. In Figure 4b, the absence of the values for the control group in these graphs make the statistical comparison and the reported significance levels quite obscure, and it would be beneficial to add them.

Thank you for this comment and opportunity to clarify this apparent confusion. In the ischemia-reperfusion studies, the baseline (pre-ischemia) values of parameters serve as the control. These parameters are expected to remain constant in control hearts for the duration of this protocol (10 mins) based on previous studies by us and others. Therefore, all comparisons in Figure 4b are made with respect to the baseline or t=0 mins value. This is now further explained in the manuscript in lines 196-199.

“Significant modulation of each parameter in each heart was determined with respect to the baseline (pre-ischemic, t=0) value from that same heart. Since no changes in any of the measured parameters are expected over the short duration (10 mins) of this protocol without any external perturbations, baseline values serve as control.”

8. As a general comment, NADH fluorescence is always reported as “normalized”, however the reference value of the normalization is not clear, since the meaning of “control” arguably changes in each of the performed experiments.

NADH normalization is illustrated in the new Figure 1d panel and further explained in lines 107-110 and 485-487. Briefly, NADH intensity changes from experiment to experiment depending on heart position, excitation light position and intensity and other factors. Therefore, during an experiment the heart was not moved after the first recording. Then, NADH intensity from each subsequent recording was normalized to this first recording. The first recording in the drug testing studies is the Control condition at 200 ms BCL. The first recording in the ischemia protocol is the Baseline condition at 200 ms BCL. This allows for the determination of modulation of NADH intensity with different experimental treatments.

“Since this parameter can vary between hearts depending on experimental conditions, all measurements from a given heart were normalized to the first recording from that same heart (Control at 200 ms pacing rate for drug testing protocol and Baseline at 200 ms pacing rate for ischemia protocol).”

“NADH intensity in each heart was normalized to the first measured NADH intensity value from that particular heart in order to avoid interexperimental variability.”

9. The acronym BCL is never explained.

We have now explained the acronym for BCL which is basic cycle length on Line 160.

10. Concerning the Online Methods sections, while I appreciate the effort to deliver a detailed GitHub repository for the data analysis code and optical setup, I would personally suggest to adopt a different platform, such as Zenodo, so that it would have a DOI and a future reader could see the state of the methods at the time of writing, and linking the GitHub repository to give the opportunity of following the latest developments.

We thank the reviewer for this helpful suggestion! The Matlab code has been uploaded to Github and linked to Zenodo. The DOI is now included in the manuscript on line 103-104 and 512).

DOI: 10.5281/zenodo.5784023.

URL: Zenodo: <https://doi.org/10.5281/zenodo.5784024>

Github: <https://github.com/optocardiography/Rhythm-3.0>.

While an emphasis on simultaneous acquisition is given throughout the text, is not clear from this section how this is implemented. How the cameras are triggered for acquisition, and the acquisition parameters adopted are left untold, and even a brief mention is necessary to understand how the authors operated. While I understand the care that the authors put into compiling the optical component list, it is still necessary to use proper terminology. The kind of excitation light source, the presence of condensers to illuminate the sample, are simply not reported. Furthermore, why the authors choose to disclose some details in the text, such as the cutoff wavelengths of the dichroic mirrors or the operating range of the bandpass filters, but withhold some (such as the focal lengths of the lenses in front of the cameras and the resulting effective magnification or pixel size) is unclear. Finally, while all the designs are freely released in an “open source” way (despite no license information being present in the GitHub repository), the necessary montage instructions are not provided. How the software mentioned helps the spatial alignment of the cameras is also not clear. More details on the treatment protocols of the heart samples would also be beneficial. For example, it has been very difficult for the present reviewer to retrieve spectral information concerning the RH237 dye, without knowing the supplier.

We thank the reviewer for these important comments and the opportunity to provide a more detailed Methods section. We have addressed all of the reviewers queries above and also added additional information throughout the Methods section to clearly detail all the aspects of the optical mapping system, drugs and reagents used. This additional information is now available between lines 377 – 487.

11. In line 403, the “very poor quality” remark is somewhat mysterious. The authors should elaborate on what does this mean. Is it due to phototoxicity, fluorescence bleaching, or some other factor?

We thank the reviewer for this comment. Poor quality signals refer to those with low signal-to-noise ratios. Without perfusion, such deterioration of signal quality is expected and can affect the accurate measurement of some of the parameters in this study. This is now explained in lines 456-459.

“Signal quality deteriorates during periods of ischemia due to poor or no perfusion of the tissue and quickly recovers during reperfusion. Poor quality signals are defined as those with low signal-to-noise ratios that would not allow for the accurate measurements of the parameters in this study.”

12. Version 3.0 of the Rhythm software is not online, please rectify.

Thank you! Rhythm 3.0 is now available online at github.com/optocardiography/Rhythm-3.0 and has been linked to Zenodo as well as the reviewer recommended.

13. In line 428, the authors should state clearly NADH fluorescence, instead of just NADH

Thank you! This has been corrected. New line 484.

14. The “Statistics” section is definitely incomplete and not sufficiently elaborated, as already stated above. Please expand with all the necessary details to replicate the statistical analysis performed by either the authors or their software.

The statistical tests performed have now been explained with all details necessary to replicate the analysis. This information can be found on new lines 488-508.

Reviewer #2:

This is a very nice methods paper, with good rationale and results. It is well written and code-sharing (for analysis and 3D printing design) is commendable. Also it shows a path towards more complex assessment of drug effects, which is important, as current assessment techniques do not provide a complete enough picture. I also like that the techniques described do not appear to be massively complex, so this is something that can be easily picked up by other labs.

We thank the reviewer for these positive comments.

Questions/comments:

1) The spectral overlap between RH-237 and Rhod-2 is unpleasant, but I believe most optical mapping labs using the dye combination found a way of working with it. However, the emission spectrum of the NADH autofluorescence makes me wonder about the overlap with Rhod-2. Could that affect features like diastolic Ca assessment? (I know that Rhod-2 is not ratiometric, but e.g. in ischemia one can compare fluorescence baseline before/after nevertheless). Would it make sense to set the illumination sources in a way that a part of the field of view would report V and Ca signals, with the other part reporting NADH autofluorescence? (these then obviously wouldn't be colocalized, but I'm not sure that's a critical limitation). Or could a focused beam be used to evoke NADH autofluorescence in a small part of the field of view, allowing the rest of the FOV to not suffer from the possible spectral overlap?

We thank the reviewer for these important questions, which are indeed critically important in designing multiparametric systems. We will address them in two parts:

A) Spectral overlap between dyes

There is indeed a spectral overlap in the emission fluorescence between RH-237 and Rhod-2, similar to that of NADH and Rhod-2. The selection of appropriate optical filters (Figure 1c) can minimize the effect caused by overlap. For example, based on the filters used in our system, only a small percent of RH237 signals overlapped with the Rhod-2 AM signals as shown in the figure below. Such a low level of overlap would not significantly affect the measurement of the parameters mentioned in this study. However, as the reviewer mentions, it is possible that it could affect intensity measurements such as those made to determine changes in diastolic calcium. In such circumstances, using a split field of view may be better option. Similar idea applies to the overlap of NADH and Rhod-2.

The above figure is from 'AAT Bioquest Company' with caption added. <https://www.aatbio.com/fluorescence-excitation-emission-spectrum-graph-viewer/oe7loT23>

We now discuss the spectral overlap of the dyes in our paper on lines 408 – 410.

“With this optical system, the amount of emission spectral overlap between NADH and Rhod2-AM as well as between Rhod2-AM and RH237 can be reduced and the interference from a different channel is minimized.”

B) Splitting the FOV into separate sections for different parameters.

The suggested split FOV optical system or recording of dual signals by recording different parameters in alternating frames can provide means to accomplish zero overlap between the different optical channels as previously described [PMIDs: 31768502, 22876327]. However, it would not allow for the recording of spatially and temporally co-registered signals as described in our manuscript. Spatially and temporally aligned recording is important for several reasons, probably the most important being spatial heterogeneity in cardiac electrophysiology.

We now revise our manuscript to state the purpose of our study more clearly in lines 68 – 71.

“In this study, we report for the first time, a spatially and temporally co-registered triple-parametric optical mapping system that incorporates three cameras to simultaneously capture NADH, Vm, and Ca²⁺ signals from the same field of view.”

2) Minor question – was the solution in the Langendorff prep recycled? (I find the continuous perfusion with diluted dyes via recycling improves signal quality and dye longevity)

Thank you for this comment. We did not recirculate the Tyrode's solution in our system because for the mouse heart we use a flow rate of 1 – 2 ml/min to maintain a coronary pressure of ~80 mmHg. At this flow rate, the total volume of the perfusate used for each experiment is very small (<150 ml). The reviewer is right that recirculation may offer improved signal quality due to continuous perfusion with the dye. However, we did not have issues with signal quality in our study (with the exception of ischemia which is expected). On the other hand, one advantage of not recirculating the perfusate is that the accumulation of metabolic waste products in the perfusate can be prevented. In addition, pharmacological studies require perfusion of different concentrations of drugs, which cannot be recirculated.

3) The statistical analysis contains aspects that are not ideal. Despite my questions/comments, I do appreciate that the concrete results are not the central point of this paper, so I do not think it is critical what exactly is significant in this paper.

a. Why was ANOVA used? I did not see the p-values reported or commented. One problem is that ANOVA is highly sensitive to the assumption of equal variance (here between different bcls) – and while I cannot be absolutely sure without seeing the raw data, this looks almost certainly violated (e.g. 2c, CV_T or V-Ca delay). The second issue is that ANOVA treats distinct bcls as independent – which they are not – they are ordered. ANOVA will give the same p-value for different reordering of your x-axis, while the biological interpretations of such data would be often different.

Perhaps the most clear example is dose-response experiments, which is another example of ordered x-axis where people sometimes use ANOVA. If, for four increasing doses, the feature of interest takes mean values such as 0, 1, 2, 3 (with some sensible variance), it can be read as a straightforward dose-dependent effect. However, if the means are ordered as 2, 1, 3, 0, it looks rather like artefact or some very nontrivial dose-dependence. Yet ANOVA would give you the same p-value for these two scenarios.

For this reason, it is often recommended to do regression analysis instead. Your data are often relatively linear, so linear regression could be used (or you could fit an exponential curve in the cases of restitution-like curves). E.g. in Fig 2c/NADH, you could take, for each heart, $x = \text{bcls}$, $y =$ the difference between blebbistatin and control at the bcl, fit a line to this – do this for all the hearts – and then do a t-test (or another test if assumptions not holding) on the offset or slope across the fitted lines (in 2c, most subplots would likely show a difference in offset, rather than in slope). You can find a bit more discussion of the ANOVA on dose-dependence (or bcl, which is similar case of ordered x-axis) here:

https://www.graphpad.com/guides/prism/latest/statistics/stat_ga_multiple_comparisons_after_.htm and here: https://www.graphpad.com/guides/prism/latest/statistics/stat_posttesttrend.htm

This is an excellent point that the reviewer raises. Thank you very much! We especially thank the reviewer for pointing us to these useful resources. We have now applied regression analysis to the restitution data to determine statistical difference between treatments. All significant p values are also now reported in the manuscript. The Statistics section in the manuscript has also been significantly modified to include additional details of the analysis. Lines 488 – 508.

“Statistics & Reproducibility: All data are reported as mean \pm standard error of the mean (SEM). A sample size of 5 hearts was used for all groups, drug treatment and ischemia protocols. An alpha level of 0.05 was used in all tests.

Least-squares regression analysis was performed on the restitution data, extra sum-of-squares F test for the best fit parameters from the regression analysis and two-tailed, paired t-tests for all other data. Since multiple statistical tests were performed on this data set (10 parameters, 5 treatments), Benjamini- Hochberg correction was applied with a false discovery rate of 20%. Shapiro-Wilk test was applied to test for data normality of the raw data values as well as the best-fit parameters of the regression analysis and >95% data sets passed the normality test. However, the small sample size of this data set may be a limitation.

For the restitution data (Figures 2 and 3), regression analysis was chosen due to the ordered nature of the independent variable (BCL). For parameters that exhibited restitution property (CV_T , CV_L , AR, APD_{80} , $CaTD_{80}$, $Ca \tau$ and V_m -Ca Delay), an exponential plateau model was used and for others a simple linear regression model was used. The parameters of the best fit were compared between groups (Control vs Blebbistatin, Blebbistatin vs 4-AP, Blebbistatin vs Verapamil) using the extra sum-of-squares F test.

For the summary data in Figure 2d, 3a and 4, paired two-tailed, t-tests were performed because it assumes normal distribution of data and does not assume equal variance. In the case of figures 2d, 3a and 4c, percent change in each parameter by a given drug (Blebbistatin, 4-AP and Verapamil) versus Control was compared. In Figure 4b, statistical tests compared change in each parameter at a given time point versus Baseline ($t=0$). ”

b. I don't think there is a problem just using paired t-tests with a correction (although mind that the same issue with ordering of the x axis being ignored persists – you are losing information by treating the bcls as independent). However, in this case, I would wonder about the assumption of normality – there are cases where the variance is quite uneven for a bcl at different conditions, so that might easily translate into non-normal distribution of the differences. It would be good to include supplementary plots of the distributions of the differences entering the t-tests (or just tables with data underlying the bcl-vs-feature plots so that curious readers can check themselves). I think visual inspection here is more useful than a normality test such as Shapiro-Wilk, which is highly underpowered on $n=5$, so even if it says the data are not significantly abnormal, it does not mean much.

We thank the reviewer for these suggestions. We have now provided the entire data set from the drug treatment and ischemia protocols in Supplemental Tables 1 and 2, respectively. We also provide below a few of the frequency distribution plots for review purposes. Additionally, we have

performed the Shapiro-Wilk test to test for normality. But as the reviewer points out, it may be underpowered and this is mentioned in the manuscript on lines 496-497.

c. A related point – please correct the wording in phrases such as “we report no change in APD in mouse hearts treated with 1 uM verapamil” – this is simply not true, there is a difference. Yes, it is often not significant, but there is almost always a difference. This is particularly important in a study with such low n’s, and possibly sometimes violated test assumptions – the fact that something was not significant does not mean much, and definitely not that there was “no difference”. See e.g., Arnheim et al. 2019 (<https://pubmed.ncbi.nlm.nih.gov/30894741/>) for how much pain and distress the phrase “there was no difference” causes to statisticians. Here I’d just write something like “there was no statistically significant difference”/“verapamil did not induce a statistically significant changes in calcium reuptake” etc. – and clarify somewhere (e.g. Limitations) that the study has very low statistical power with n=5 and a large number of tests carried out with correction, so real effects were likely to be missed.

We appreciate the reviewer pointing out this nuance in wording and how it can be interpreted. We have now edited the manuscript to address statistical significance as recommended. This was corrected at the following lines.

Line 204: *“Changes in AR and Ca²⁺ RT were not statistically significant during the 5 min ischemic protocol.”*

Lines 299 – 300: “In this study, we report no statistically significant changes in APD in mouse hearts treated with 1 μ M verapamil.”

Lines 303 – 304: “In line with these findings, we report no statistically significant changes in V_m RT and CV in mouse hearts treated with verapamil.”

Lines 310-312: “Lastly, we also report here that verapamil did not induce any significant changes in calcium reuptake as indicated by no changes in CaTD and Ca_r parameters.”

d. Bonferroni correction is very conservative and particularly on n=5, it is very likely to make you miss a lot of genuine effects. Did you consider using the Benjamini-Hochberg correction instead?

We thank the reviewer for this helpful comment. We have now applied Benjamini-Hochberg correction to our data to account for multiple comparisons.

e. Why was s.e.m. used instead of standard deviation? Stdev is a generally preferable indicator of data spread, with s.e.m. often making the variability look smaller than it is. This is particularly critical with uneven group size, which is not a problem in your paper, but still, it does not give a great picture of the data variability. See e.g. <https://diabetes.diabetesjournals.org/content/62/8/e15> or <https://journals.plos.org/plosone/article?id=10.1371/journal.pone.0110364>. I appreciate that s.e.m. also has its place in statistical analysis, but in cases where you report it, it appears that a standard deviation would be more appropriate.

We agree with the reviewer that standard deviation is more appropriate. We have now included a table with the entire data set represented as mean \pm standard deviation. However, in the figures, due to the large amount of data and multiple, sometime overlapping curves, we chose to use SEM in order to provide clearer visualization of the data and avoid confusion.

f. Your text is not always conforming to the statistical guidelines of the journal <https://www.nature.com/commsbio/submit/submission-guidelines#statistical-guidelines> – e.g. accompanying significance statements with p-values.

We thank the reviewer for pointing this out. We have redone the statistical tests according to the requirements of the journal and the reviewer’s suggestion and explained this in detail in the Statistics section (Lines 488-508). All journal guidelines have now been addressed.

Reviewers' comments:

Reviewer #1 (Remarks to the Author):

The revised version of the manuscript submitted by the authors shows considerable effort and willingness to take on board many of the suggestions by both reviewers, which I personally greatly appreciate. Probably due to the significant additions, there are still a few points which should be addressed, in order to complete the picture offered by the authors and offer sufficient details to the readers to fully appreciate the approach herein presented, as well as improve the article in general. I will be addressing remaining issues with the paper in the order they are presented in the rebuttal letter.

1) In Figure 1d, it seems a "(Right)" indication is missing alongside (Left) and (Middle).

2) The addition of the regression analysis, suggested by the other reviewer, is certainly worthwhile. However, the parameters reported in the Supplemental Table, as well as the description of the methodology, leave many obscure points:

- How are the regression performed? Was an ad-hoc code generated, or an external software employed? This is non-trivial, especially in the case of the in-house code, as it should also be available, or at least better described, to evaluate its appropriateness to the case;
- The mathematical formulation of the models employed is missing, together with the specification of the number of floating parameters;
- There is no sign of a goodness-of-fit metric (e.g., R-square tests, chi-square). These would be especially important in the case of Vm-Ca²⁺ delay, as it visually appear that no clear correlation can be established between the two variables;
- A more exact Supplemental Table should include more detail on which parameters are reported, and what are their units.

3) The caption of Figure 4b is somewhat not sufficient to understand the meaning of the figure by itself. The mention that "the modulation of each parameter in each heart was determined with respect to the baseline" should also be included there.

4) Finally, the description of the optical setup and the presentation of the 3D designs should be readjusted for full clarity, although I appreciate the additional details already put in place to give a clearer picture of the instrument. Points to be addressed are:

- Licensing information on the Github repository with the .stl files is still missing. This is very important, as it currently is not clear under what circumstances the files can be used (either by companies for commercial purposes, or other scientists) and there is actually no obligation to attribute the original work to the authors (this is no actual legal advice, I am just speaking out of my best knowledge). I suggest then that a clear license file is added, such as the authors have already done in their other repositories;
- Somewhere in the text should be mentioned that this system is designed specifically to use with the SciMedia equipment present in the lab;
- The terms working distance and focal length are not exactly interchangeable;
- The wording to describe the lenses is somewhat confusing, particularly when the mention "(projection lens, same as above)" is used. A better reference would be "... focused by a second planapo 1X objective on the imaging sensor", or something similar.

I look forward to receive an amended version of the manuscript, and I thank the authors for their work.

Reviewer #2 (Remarks to the Author):

The authors have addressed my points very well and I was happy to read they found the comments useful!

Tiny points only:

1) On line 504, "paired two-tailed, t-tests were performed..."

- I think the comma can be removed

- I'm not too sure about the wording of the justification for the use of the test - maybe you could just say these tests were performed and skip the part "because it assumes..."? The paired test is clearly suitable, given the paired nature of the data (as long as the pair differences are reasonably

normal). I.e., the normality is more a requirement, than a justification for use. Also, for the "and does not assume equal variance" - that is indeed a feature of the paired test, but it's not a reason to use it necessarily. It could be read as you are selecting a paired over unpaired test, given that it doesn't require equal variance - but the decision here is because of the paired, rather than unpaired experimental design.

Was the Benjamini-Hochberg applied for the different comparisons within figures 2d, 3a and 4c? It seems like it would be suitable. If yes, it would be worth clarifying it in figure legends for maximum clarity.

Response to Reviewers

We thank the reviewers for taking the time to review our revised manuscript and for providing us with additional constructive feedback. We believe that this opportunity to revise has further improved our manuscript. We have addressed all the reviewers concerns and made necessary edits in the manuscript as suggested. Below, we describe how each concern was addressed in the revised manuscript.

Reviewer #1:

The revised version of the manuscript submitted by the authors shows considerable effort and willingness to take on board many of the suggestions by both reviewers, which I personally greatly appreciate. Probably due to the significant additions, there are still a few points which should be addressed, in order to complete the picture offered by the authors and offer sufficient details to the readers to fully appreciate the approach herein presented, as well as improve the article in general. I will be addressing remaining issues with the paper in the order they are presented in the rebuttal letter.

1) In Figure 1d, it seems a “(Right)” indication is missing alongside (Left) and (Middle).

We thank the reviewer for catching this missing text. We have now labelled the three panels at the bottom of Figure 1 as panels d, e and f to avoid apparent confusion. The corresponding figure legend and manuscript text has also been updated on lines 671-675 and 105, respectively.

Line 105: *“The following 10 parameters were measured from the optical recordings and an illustration of each parameter definition is included in Figure 1d-f.”*

Lines 679 – 684: *“Illustration of the different parameters measured using the three optical signals. **d**) NADH intensity was determined and each intensity value from a given heart was normalized to the NADH intensity from the first recording from that heart. **e**) Definition of parameters measured from the depolarization phase of the V_m signal or the calcium release phase of the Ca^{2+} signal. **f**) Definition of the parameters measured from the repolarization phase of the V_m signal or the calcium reuptake phase of the Ca^{2+} signal.”*

2) The addition of the regression analysis, suggested by the other reviewer, is certainly worthwhile. However, the parameters reported in the Supplemental Table, as well as the description of the methodology, leave many obscure points:

- How are the regression performed? Was an ad-hoc code generated, or an external software employed? This is non-trivial, especially in the case of the in-house code, as it should also be available, or at least better described, to evaluate its appropriateness to the case;

- The mathematical formulation of the models employed is missing, together with the specification of the number of floating parameters;

- There is no sign of a goodness-of-fit metric (e.g., R-square tests, chi-square). These would be especially important in the case of V_m - Ca^{2+} delay, as it visually appear that no clear correlation can be established between the two variables;

- A more exact Supplemental Table should include more detail on which parameters are reported, and what are their units.

We thank the reviewer for bringing to our attention the missing information on the regression analysis. We have now included all the information requested by the reviewer including software used, model equations and metric of good fit in the Statistics section of the manuscript on lines 494-510.

As for V_m-Ca delay, the reviewer correctly points out that there is no clear pattern and even though we tried fitting multiple models, the r-squared value was still low. However, third order polynomial model seemed to give the best fit. We also generated the residual plot (residual vs dependent variable) and the residuals were randomly scattered around 0. This is also now mentioned in the manuscript.

“Regression analysis was performed on the restitution data using Graphpad Prism Version 9.3.1 software. Non-linear regression analysis was performed using the least squares regression fitting method. For parameters that exhibited restitution property (CV_T, CV_L, AR, APD₈₀, CaTD₈₀, and Ca τ), an exponential plateau model ($Y = Y_M - (Y_M - Y_0) e^{-kX}$) was used. For V_m-Ca delay parameter, third order polynomial model ($Y = B_0 + B_1X + B_2X^2 + B_3X^3$) was used. For all others a simple linear regression model ($Y = a + bX$) was used. All other parameters were left at default settings in this software. Good model fit was determined by r-squared value > 0.5 (except V_m-Ca delay during blebbistatin treatment) and by confirming the random nature of the residual plots.

The parameters of the best fit were compared between groups (Control vs Blebbistatin, Blebbistatin vs 4-AP, Blebbistatin vs Verapamil) using the extra sum-of-squares F test. Two-tailed paired t-tests were performed for all other data. Since multiple statistical tests were performed on this data set (10 parameters, 5 treatments), Benjamini- Hochberg correction was applied with a false discovery rate of 20%. Shapiro-Wilk test was applied to test for data normality of the raw data values as well as the best-fit parameters of the regression analysis and >95% data sets passed the normality test. However, the small sample size of this data set may be a limitation.”

We have also added the description of the parameters along with their units in the table legends for the Supplemental Tables 1 and 2 as the reviewer suggested.

“The values reported in this table include V_m RT (ms): action potential upstroke rise time, APD₈₀ (ms): action potential duration at 80% repolarization, CV_T (m/s): transverse conduction velocity, CV_L (m/s): longitudinal conduction velocity, AR: anisotropic ratio, Ca²⁺ RT (ms): calcium transient upstroke rise time, CaTD₈₀ (ms): calcium transient duration at 80% reuptake, Caτ (ms): calcium transient decay time constant, V_m-Ca delay (ms): time delay between upstroke of action potential and calcium transient, NADH: normalized NADH fluorescence intensity.”

3) The caption of Figure 4b is somewhat not sufficient to understand the meaning of the figure by itself. The mention that “the modulation of each parameter in each heart was determined with respect to the baseline” should also be included there.

We thank the reviewer for this excellent suggestion to further clarify the data normalization used in Figure 4. We have now added the following sentence in the Figure 4 legend on lines 752-754, as the reviewer suggested.

“The values reported are normalized to baseline ($t=0$) for each heart in order to determine modulation of each parameter in each heart with respect to its own baseline condition.”

4) Finally, the description of the optical setup and the presentation of the 3D designs should be readjusted for full clarity, although I appreciate the additional details already put in place to give a clearer picture of the instrument. Points to be addressed are:

- Licensing information on the Github repository with the .stl files is still missing. This is very important, as it currently is not clear under what circumstances the files can be used (either by companies for commercial purposes, or other scientists) and there is actually no obligation to attribute the original work to the authors (this is no actual legal advice, I am just speaking out of my best knowledge). I suggest then that a clear license file is added, such as the authors have already done in their other repositories;

We thank the reviewer for bringing this to our attention. We have now uploaded a license file to the STL files repository on Github. We understand that attribution is not enforceable, but our goal is to increase availability and impact of our technology as much as possible. In our experience, for the most part users of our code and technology cite the source.

- Somewhere in the text should be mentioned that this system is designed specifically to use with the SciMedia equipment present in the lab;

Thank you for bringing this to our attention. Yes, the 3D printed parts that attach to the camera and the software were designed to work with SciMedia equipment and data formats. All other parts can be used in conjunction with other camera types. We have now stated this in the manuscript on lines 389-390 and 465-467.

Lines 389 – 390: *“The camera cages were specifically designed for SciMedia cameras but the other system components can be used with optical mapping systems from other sources.”*

Lines 465 – 467: *“Rhythm software was written to analyze data formats generated by SciMedia systems but can be modified to analyze other data formats.”*

- The terms working distance and focal length are not exactly interchangeable;

We agree with the reviewer that the use of these two terms side by side in the manuscript could cause confusion. We have corrected the manuscript text and removed the term focal length accordingly on line 397 – 398.

“The emitted light is collected by an infinity-corrected, planapo 1X lens (objective lens, SciMedia) with a working distance of 61.5 mm.”

- The wording to describe the lenses is somewhat confusing, particularly when the mention “(projection lens, same as above)” is used. A better reference would be “... focused by a second planapo 1X objective on the imaging sensor”, or something similar.

We thank the reviewer for this opportunity to clarify this sentence. We have edited the sentence on lines 400 – 402, as suggested. We have left the term “projection lens” in parenthesis to distinguish between the objective and projection lenses when it is later described in the text.

“Light at different wavelengths were separated using dichroic mirrors and filters as illustrated in Figure 1c and passed through a second planapo 1X lens (projection lens, SciMedia) before being recorded using the CMOS cameras.”

I look forward to receive an amended version of the manuscript, and I thank the authors for their work.

We thank the reviewer for once again providing us with valuable feedback which allowed us to further improve our manuscript.

Reviewer #2:

The authors have addressed my points very well and I was happy to read they found the comments useful!

Tiny points only:

On line 504, "paired two-tailed, t-tests were performed..." - I think the comma can be removed

We thank the reviewer for this suggestion. We have removed the comma in this sentence. New line 511.

“For the summary data in Figure 2d, 3a and 4, paired two-tailed t-tests were performed.”

- I'm not too sure about the wording of the justification for the use of the test - maybe you could just say these tests were performed and skip the part "because it assumes..."? The paired test is clearly suitable, given the paired nature of the data (as long as the pair differences are reasonably normal). I.e., the normality is more a requirement, than a justification for use. Also, for the "and does not assume equal variance" - that is indeed a feature of the paired test, but it's not a reason to use it necessarily. It could be read as you are selecting a paired over unpaired test, given that it doesn't require equal variance - but the decision here is because of the paired, rather than unpaired experimental design.

We agree with the reviewer that this justification can cause further confusion as the reviewer suggests. So we have now removed this statement from the manuscript. Thank you for the opportunity to clarify this point in line 511.

“For the summary data in Figure 2d, 3a and 4, paired two-tailed t-tests were performed.”

Was the Benjamini-Hochberg applied for the different comparisons within figures 2d, 3a and 4c?

It seems like it would be suitable. If yes, it would be worth clarifying it in figure legends for maximum clarity.

We thank the reviewer for this suggestion to provide further clarity in our statistical analysis. Yes, indeed, we did apply the Benjamini-Hochberg correction to data summary data presented in the TPP graphs as well and this is now explained in the legends for Figures 2, 3 and 4.

“Benjamini-Hochberg correction was applied to account for multiple comparisons.”